

# Sensitivity analysis of DSD retrievals from polarimetric radar in stratiform rain based on $\mu$-$\Lambda$ relationship

Christos Gatidis[1], Marc Schleiss[1], and Christine Unal[1]

[1]Department of Geoscience and Remote Sensing, Delft University of Technology, Delft, The Netherlands

**Correspondence:** Christos Gatidis (C.Gatidis@tudelft.nl)

**Abstract.** Raindrop size distributions (DSD) play a crucial role in quantitative rainfall estimation using weather radar. Thanks to dual-polarization capabilities, crucial information about the DSD in a given volume of air can be retrieved. One popular retrieval method assumes that the DSD can be modeled by a constrained gamma distribution in which the shape ($\mu$) and rate ($\Lambda$) parameters are linked together by a deterministic relationship. In the literature, $\mu$-$\Lambda$ relationships are often taken for granted and applied without much critical discussion. In this study, we take another look at this important issue by conducting a detailed analysis of $\mu$-$\Lambda$ relations in stratiform rain and quantifying the accuracy of the associated DSD retrievals. Crucial aspects of our research include the sensitivity of $\mu$-$\Lambda$ relations to the temporal aggregation scale, drop concentration, inter-event variability and adequacy of the gamma distribution model. Our results show that $\mu$-$\Lambda$ relationships in stratiform rain are surprisingly robust to the choice of the sampling resolution, sample size and adequacy of the gamma model. Overall, the retrieved DSDs are in a rather decent agreement with ground observations (correlation coefficient of 0.57 and 0.74 for $\mu$ and $D_m$). The main sources of errors and uncertainty during the retrievals are calibration offsets in reflectivity ($Z_{hh}$) and differential reflectivity ($Z_{dr}$). Measurement noise and differences in scale between radar and disdrometers also play a minor role. The most difficult to retrieve parameter remains the raindrop concentration ($N_T$), which can be off by several orders of magnitude. By removing problematic $Z_{hh}$/$Z_{dr}$ pairs, the correlation coefficient for the retrieved $N_T$ values increases from 0.12 to 0.24, however even after the careful data filtering the accuracy of the retrieved values remains low.



# 1 Introduction

Understanding the natural variability of raindrop size distributions (DSD) is crucial for radar remote sensing applications and microphysical parametrizations in numerical weather prediction models (e.g. Thompson et al. (2004)). Most precipitation-
related quantities (e.g. rain rate, mean drop diameter, number concentration, fall velocity or liquid water content) directly depend on the DSD. Similarly, most radar observables (e.g. $Z_{hh}$, $Z_{dr}$) are weighted moments of the DSD. For these reasons, DSD retrieval methods play a central role in numerous weather radar studies.

Efforts to improve quantitative rainfall estimates by retrieving information about DSDs from radar and satellite observations have captured a great deal of interest in the meteorological community, especially after the introduction of polarimetric
weather radar (Seliga and Bringi, 1976). Retrievals based on the reflectivity factor at horizontal polarization ($Z_{hh}$), differential reflectivity ($Z_{dr}$) and specific differential phase ($K_{dp}$) are the most common choices, because of their natural link to raindrop concentrations, sizes and shapes.

According to literature, DSDs can be parameterized in the form of relatively simple models such as a gamma distribution with three parameters $\mu$, $\Lambda$ and $N_0$ representing the shape, scale and concentration respectively. Algorithms for DSD retrievals take
advantage of different relationships between radar observables and the three parameters of the gamma. Three main categories of retrieval methods can be distinguished: the first one consists of methods that use two radar observations $Z_{hh}$ and $Z_{dr}$, and a constrained relationship between $\mu$ and $\Lambda$ (Zhang et al., 2001, 2003) or $N_0$ and $\mu$ (Ulbrich, 1983). The second category proposed by Bringi et al. (2002) and Gorgucci et al. (2002) uses three radar observables $Z_{hh}$, $Z_{dr}$ and $K_{dp}$. However, this method is known to be very sensitive to noise in $K_{dp}$ estimates. To reduce the uncertainty, the differential phase needs to
be filtered and down-sampled, which limits the accuracy and spatial resolution of the retrievals. The last category consists of various retrieval techniques that require special types of radars or measurements, such as double frequency (Rose and Chandrasekar, 2006), triple frequency (Mróz et al., 2020) and/or Doppler power spectra (Unal, 2015). In this paper, only the first category will be discussed.

The main challenges when retrieving DSDs from $Z_{hh}$ and $Z_{dr}$ are the choice of the $N_0$-$\mu$ or $\mu$-$\Lambda$ relationship and its validity
across different rain types and spatial and temporal aggregation scales. In the literature, $\mu$-$\Lambda$ relationships are often taken for granted or transferred from one location or scale to another without much critical discussion. And while some studies have documented large differences in relationships across rain types (e.g., stratiform vs convective), little remains known about the sensitivity of $\mu$-$\Lambda$ relationships to the temporal sampling resolution of the disdrometer data used to infer them or the validity of the gamma assumption. Another important issue concerns the fact that the disdrometer data used to define $\mu$-$\Lambda$ relationships
correspond to much smaller sampling volumes than the radar measurements on which they are applied. Therefore, it might be necessary to first apply a statistical transformation to the radar data before retrieving DSDs based on $\mu$-$\Lambda$ relationships or, equivalently, modify the $\mu$-$\Lambda$ relation to account for the difference in scale. Finally, one last issue that is often overlooked is that radar measurements are likely to contain systematic errors in the form of calibration offsets in $Z_{hh}$ and $Z_{dr}$. The latter can induce large biases in the retrieved DSDs, especially in light rain with low $Z_{dr}$ and small signal to noise ratio. Despite the





radar community's best efforts, calibration errors remain hard to assess quantitatively due to the lack of a trusted reference. As a result, the quality of the retrieved DSDs strongly depend on the error characteristics of a radar and how well it is calibrated.

In this paper, we perform a detailed analysis of the sensitivity of DSD retrievals from polarimetric radar to various error sources such as the validity of the $\mu$-$\Lambda$ relationship and its sensitivity to the temporal sampling resolution, inter-event variability, changes in number concentrations and adequacy of the gamma distribution model. We also examine the sensitivity of the
retrievals to measurement biases in $Z_{hh}$ and potential biases in $Z_{dr}$ due to differences in measurement scale. We illustrate the importance of all these issues by retrieving DSDs during several episodes of light to moderate stratiform rain in Cabauw, the Netherlands and indirectly validating our retrievals by comparing them to disdrometer observations on the ground. The main focus is not on optimizing the DSD retrieval algorithm but on understanding its sensitivity to potential sources of errors, either directly linked to the radar measurements or indirectly through the critical modeling assumptions behind the method.

This paper is organized as follows. In Section 2, the data used are introduced. In Section 3, the methodology is presented. In Section 4, the main results for the $\mu$-$\Lambda$ relationship analysis are shown, followed by the sensitivity analysis on the DSD retrievals in Section 5. Finally, the conclusions are provided in Section 6.

## 2  Data

The data used in this study were collected in the Netherlands during the ACCEPT (Analysis of the Composition of Clouds
with Extended Polarization Techniques) campaign between October and November 2014. During this campaign, a variety of different in-situ and remote sensing measurements were collected at the CESAR (Cabauw Experimental Site for Atmospheric Research) observatory.

### 2.1  The disdrometer data

The ground DSD spectra used for calibration and validation were collected by a PARSIVEL[2] (PARticle SIze and VELocity)
optical disdrometer. The working principle, strengths and limitations of the PARSIVEL[2] have already been discussed in great depth in previous studies and will not be repeated here (Löffler-Mang and Joss, 2000; Tokay et al., 2014; Battaglia et al., 2010; Thurai et al., 2011). The raw DSD data consist of particle counts across 32 non-uniformly spaced diameter classes ranging from 0 to 25 mm with a sampling resolution of 30 seconds. From the raw DSD, integrated quantities such as rainfall rate (R) and radar equivalent reflectivity factor (Z) can be derived (Bringi and Chandrasekar, 2001; Thurai and Bringi, 2008). The
disdrometer measurements were used to fit gamma DSD models and derive constrained relations between $\mu$ and $\Lambda$ parameters at different temporal resolutions, which is necessary for retrieving DSDs from polarimetric radar measurements. At the same time, the disdrometer measurements were also used to (indirectly) validate the radar retrievals and study their consistency over time and across different events.



## 2.2 Radar data

The radar data used to perform the DSD retrievals were collected by TU Delft's polarimetric S-band ($\lambda$=9.1 cm) FMCW radar TARA (Transportable Atmospheric RAdar; Heijnen et al., 2000) in Cabauw, the Netherlands. For this experiment, the radar antenna elevation angle of TARA was fixed at 45° with constant azimuth. The collected polarimetric radar observables include reflectivity factor at horizontal polarization ($Z_{hh}$) and differential reflectivity ($Z_{dr}$). The full specifications of TARA during the ACCEPT campaign are given in Table 1 of Pfitzenmaier et al., 2018.

In order to make the radar data comparable with the disdrometer data, all $Z_{hh}$ and $Z_{dr}$ measurements were down-sampled over successive 30 s sampling intervals. The radar and disdrometer data were then syncronized by determining the time shift that maximized the correlation coefficient between $Z_{hh}$ Parsivel and $Z_{hh}$ TARA.

Concerning the calibration of $Z_{hh}$ and $Z_{dr}$, noise measurements were performed every day to account for possible variations in range, especially at the beginning and the end of the IF-filter. Before the start of the campaign, the calibration of $Z_{dr}$ was

verified using vertical profiling of drizzle and very light rain. The resulting histograms showed a mean offset of -0.11 dB with a standard deviation of 0.05 dB. Consequently, an offset of +0.11 dB was added to the measured $Z_{dr}$ for the whole ACCEPT campaign. For the calibration of $Z_{hh}$, the transmit power was stored in the data set, and there was a near-field correction for the non-full-overlapping of the transmit and receive antenna beams using the method described in (Sekelsky and Clothiaux, 2002). However, an end-to-end calibration for $Z_{hh}$ was missing.

## 2.3 List of events


A total of 7 rain events over the whole measurement campaign were selected for further analysis. The criteria used to select events were as follows:

1. Each event must consist of predominantly stratiform rain and exhibit a clear melting layer signal in the radar data.

2. Each rain event must be at least two hours long in duration. This was deemed necessary to have enough data to fit a
reliable $\mu$-$\Lambda$ relation and compute relevant performance metrics.

3. There should be no clear sign of changes in dynamics or microphysics (Jameson and Kostinski, 2001; Gorgucci et al., 2001; Uijlenhoet et al., 2003) with no long dry periods within each event.

4. Each event must contain several $Z_{dr}$ and $Z_{hh}$ values larger than 0.1 dB and 5 dBZ respectively.

Table 1 presents a summary of the duration, rain intensity and mass-weighted mean diameter (based on the disdrometer data)
for each of the seven selected events. As can be seen, most of the events last between 120 and 150 minutes. The longest on November 3 is slightly longer than 4 hours. The low rain intensity and mass-weighted mean drop diameter values confirm that the selected events are mostly comprised of light to moderate stratiform rain. This makes sense given the criteria used to select the events and the fact that the ACCEPT campaign took place in October-November in the Netherlands, at a time when heavy convective events are rare.





For illustration purposes, one of the 7 events (E2, 11 October, 2014) is plotted in Fig. 1. As can be seen, this event mostly
consists of stratiform rain with a moderate intensity of approximately 1.8 mm h$^{-1}$ and a total duration of approximately 3 hours
between 10:30 and 13:45 UTC, including a short break between 12:45 and 12:55 UTC according to disdrometer observations
on the ground (Fig. 2). The mass-weighted mean diameter is 1.1 mm, which is typical for light stratiform rain and small
raindrop sizes. Event 2 was chosen because it has a relatively stable, well defined melting layer around 2 km height as shown
by the enhanced values of $Z_{hh}$ and $Z_{dr}$ in Fig. 1 top and bottom respectively. The event also has a relatively low horizontal
wind speed which makes it easier to compare the radar retrievals aloft with the disdrometer measurements on the ground.

## 3    Methods

### 3.1    DSD model

The model used to approximate raindrop size distributions (DSDs) in this paper is the gamma distribution proposed by Ulbrich
120  (1983):

$$N(D) = N_0 D^\mu e^{-\Lambda D} = N_T \frac{\Lambda^{\mu+1} D^\mu}{\Gamma(\mu+1)} e^{-\Lambda D}, \tag{1}$$

where N(D) is the raindrop size distribution in mm$^{-1}$m$^{-3}$, $\mu$ is the shape parameter (unitless), $\Lambda$ is the slope parameter (mm$^{-1}$),
$N_0$ is the intercept parameter (mm$^{-1-\mu}$m$^{-3}$) and $N_T$ is the total number concentration (m$^{-3}$). The advantage of $N_T$ over $N_0$
is that its unit does not depend on $\mu$. For convenience, the gamma model is reformulated in terms of the mass-weighted mean
diameter $D_m$ (mm) and the generalized intercept parameter $N_w$ (mm$^{-1}$m$^{-3}$) (Testud et al., 2001; Bringi et al., 2003) to:

$$N(D) = N_w f(\mu) \left( \frac{D}{D_m} \right)^\mu e^{-(4+\mu)\frac{D}{D_m}}, \tag{2}$$

where $f(\mu)$, $N_w$ and $D_m$ are given by:

$$f(\mu) = \frac{6}{4^4} \frac{(\mu+4)^{(\mu+4)}}{\Gamma(\mu+4)}, \tag{3}$$

$$N_w = \frac{4^4}{\pi \rho_w} \left( \frac{LWC}{D_m{}^4} \right), \tag{4}$$

$$D_m = \frac{4+\mu}{\Lambda}. \tag{5}$$




In the equations above, LWC denotes the liquid water content (in g m$^{-3}$) and $\rho_w$ is the density of liquid water ($10^{-3}$ g mm$^{-3}$). It should be mentioned that even though the gamma distribution is the most popular and widely accepted model for representing DSDs in the literature, several studies have questioned its adequacy (Gatidis et al., 2020; Thurai et al., 2019; Cugerone and De Michele, 2015), setting criteria and proposing different tools to check the gamma hypothesis on a case-by-case basis.

### 3.2 Parameter fitting

The best parameters ($\mu$, D$_m$ and N$_w$) for describing the DSDs measured by the disdrometer are obtained by applying the well-established method of moments (MoM) in Thurai et al. (2014). The value of $\mu$ is determined by testing all possible values of $\mu$ between -3 and 15 and choosing the one that minimises the cost function (CF) below:

$$\text{CF} = \sum_{i=3}^{22} |\log_{10}(N_{\text{obs}}(D_i)) - \log_{10}(N(D_i \mid \mu))|, \tag{6}$$


where D$_i$ is the center of the i$^{th}$ diameter class in the Parsivel disdrometer and N$_{\text{obs}}$(D$_i$) are the volumetric size distribution measurements for each diameter class. Note that the index i ranges from 3 to 22 because the first two diameter classes in the Parsivel are always zero and the diameter classes above 22 correspond to particles that are too large to be associated with rain.

### 3.3 DSD retrieval method

Because the gamma DSD model involves three parameters, three different radar measurements representative of three weighted moments of the DSD are required to retrieve the DSD in a given radar resolution volume. The retrieval method used in this paper is described in Zhang et al. (2001). It involves a combination of reflectivity factor at horizontal polarization (Z$_{hh}$), differential reflectivity (Z$_{dr}$), and a deterministic relationship between the DSD shape parameter ($\mu$) and slope parameter ($\Lambda$), commonly referred to as a $\mu$-$\Lambda$ relationship. The main steps of the retrieval method can be summarized as follows:

1. Impose a $\mu$-$\Lambda$ relationship $\Lambda = g(\mu)$ based on nearby disdrometer observations or literature values. In our case, a power-law relationship is used:

$$\Lambda = 0.514(\mu + 3)^{1.339} \tag{7}$$

where the prefactor and exponent were determined by combining all the data from all seven events in Table 1.

  2. Consider all possible values of $\mu$ between -3 and 15 in steps of 0.01. For each $\mu$ value, calculate Z$_{dr}$ through Eq. 8:

$$Z_{dr} = \frac{Z_{hh}}{Z_{vv}} = \frac{\int_0^{D_{max}} N(D)\sigma_{hh}(D)dD}{\int_0^{D_{max}} N(D)\sigma_{vv}(D)dD} = \frac{\int_0^{D_{max}} D^\mu e^{-g(\mu)D}\sigma_{hh}(D)dD}{\int_0^{D_{max}} D^\mu e^{-g(\mu)D}\sigma_{vv}(D)dD} = \frac{h_1(\mu)}{h_2(\mu)}, \tag{8}$$





where $\sigma_{hh}$ (mm$^2$) and $\sigma_{vv}$ (mm$^2$) are the copolar radar cross-sections of raindrops with equivolume spherical diameter D, at horizontal and vertical polarizations, respectively, and D$_{max}$ (mm) is a reasonable maximum drop diameter (e.g., 7 mm in our case). The detailed expression of the radar cross-sections can be found in Eq. 3 in (Unal, 2015).

3. Keep the $\mu$ value for which the Z$_{dr}$ value in Eq.(8) is closest to the measured Z$_{dr}$ value by the radar.

   4. Infer $N_w$ from Z$_{hh}$ in Eq. (9), where $\hat{\mu}$ is the retrieved $\mu$ value from the previous step:

$$Z_{hh} = N_w \frac{\lambda^4 f(\hat{\mu})}{\pi^5 |K_w|^2} \int\limits_0^{D_{max}} \left(\frac{D}{\hat{D}_m}\right)^{\hat{\mu}} e^{-(4+\hat{\mu})\frac{D}{\hat{D}_m}} \sigma_{hh}(D)dD, \tag{9}$$

where $\Lambda$ is the radar wavelength in mm, (i.e., 90.96 mm for TARA), $|K_w|^2$ is the dielectric factor of water and $\hat{D}_m = \frac{4+\hat{\mu}}{g(\hat{\mu})}$.

5. Retrieve $\hat{N}_T$ by integrating the retrieved DSD:

$$\hat{N}_T = \int\limits_0^{D_{max}} \hat{N}(D)dD = \int\limits_0^{D_{max}} \hat{N}_w f(\hat{\mu})\left(\frac{D}{\hat{D}_m}\right)^{\hat{\mu}} e^{-(4+\hat{\mu})\frac{D}{\hat{D}_m}} dD. \tag{10}$$

## 3.4 $\mu$-$\Lambda$ relation

The usage of a deterministic relationship between shape and scale is a convenient way to reduce the number of free parameters in radar-based DSD retrievals. Numerous studies have used and proposed constrained relationships between these two DSD parameters. The most common models are based on second-order polynomial fits, firstly introduced by Zhang et al. (2001, 2003). Since then, several other studies have proposed updated polynomial $\mu$-$\Lambda$ relationships based on either seasonal (Seela et al., 2018) or regional criteria (Chen et al., 2016). Polynomial models between $\mu$ and $\Lambda$ were also proposed for DSD retrievals using microwave link measurements (Berne and Schleiss, 2009; van Leth et al., 2020). In this study, $\mu$-$\Lambda$ relationships are modeled using a slightly different power-law model:

$\Lambda = \alpha(\mu + 3)^{\beta}$, $\hspace{5cm}$ (11)

with two coefficients $\alpha$ and $\beta$ as given in Eq. 7.

   The power-law model above was chosen mainly for mathematical reasons since it ensures that $\Lambda$ remains positive across all 185 scales and avoids the problem of having to choose between a first, second or third-order polynomial. The power law model is





also easier to justify than a parabola from a physical and mathematical point in light of the scale-invariance of DSDs under proper normalization, as pointed out by previous researchers (Torres et al., 1994; Testud et al., 2001). However, for the sake of completeness, we also examined the polynomial model during our study and concluded that it did not make a big difference from a practical point of view (i.e., it has similar goodness of fit over the considered range of $\mu$ values). Nevertheless, we

decided to use the power-law model in this study since it is more appropriate than a polynomial from a theoretical point of view.

Note that the goal of this study is not to question the validity of previous $\mu$-$\Lambda$ relationships nor optimize the parameters behind them (which depend on the used dataset) but to take a closer look at the sensitivity of the obtained fits to various underlying assumptions. Critical aspects that were investigated are whether the $\mu$-$\Lambda$ relation remains stable with respect to

different sampling resolutions, drop number concentrations, types of stratiform rain events or the validity of the gamma DSD hypothesis itself.

## 4   Analysis of $\mu$-$\Lambda$ relationship

### 4.1   Variations in $\mu$-$\Lambda$ relationship from one event to another

In the following, we analyze the variations of the $\mu$-$\Lambda$ relationships from one event to another. For this, a filter was applied

identical to Gatidis et al. (2020) and only the cases which satisfied the gamma model hypothesis were considered. In order to investigate and visualize possible differences between events, all 7 events were plotted using different colors in Fig. 3. The overall relationships by Zhang et al. (2001, 2003) were added for comparison. As can be seen in Fig. 3, most of the event-specific $\mu$-$\Lambda$ relations stay relatively close to the overall relation, except for events 2 and 6 where larger deviations for higher values of $\mu$ (i.e., $\mu$>8) are visible. For event 6, the differences can be explained by the limited range of $\mu$, with most

values remaining between 3 and 5, and only a single observation falling between 5 and 15. This limited range of variability significantly affects the reliability of the estimated $\mu$-$\Lambda$ relationship, especially for values smaller than 3 and larger than 5. For event 2, the differences can be explained by the presence of a few outliers in the upper-right part of the scatter plot, corresponding to DSDs with low number concentrations and high sampling uncertainties.

For each selected event, the sample sizes and the fitted power-law parameters $\alpha$, $\beta$ and their percentage relative differences

against the overall relation are presented in Table 1. The relative errors of the parameters depend on the characteristics of each event, with event 1 being the closest to the overall relation and event 6 exhibiting the largest differences. Incidentally, event 6 also has the smallest sample size.

The event-specific and overall $\mu$-$\Lambda$ relations are clearly different from previously proposed relations by Zhang et al. (2001, 2003). For a fixed $\mu$ value, the overall $\mu$-$\Lambda$ relation for the 7 selected events predicts higher $\Lambda$ values compared with the ones by Zhang

et al. (2001, 2003). This can be explained by the fact that $\Lambda$ is inversely proportional to the mass-weighted mean diameter and that the Zhang et al. (2001, 2003) relations were derived under different climatological conditions in Oklahoma U.S. where convective rain events with larger raindrops are more common than in the Netherlands.





Although the overall relationship might not necessarily be optimal for each individual event, our results show that it still provides a fairly good approximation of the average $\mu$-$\Lambda$ relationship across all the 7 considered events. Also, one has to keep in

mind that the low sample sizes and limited ranges for $\mu$ make it practically impossible to derive reliable and representative $\mu$-$\Lambda$ relations for each individual event. To avoid sampling issues such as those encountered in event 6, and increase the robustness of our results, all remaining sensitivity analyses and retrievals were therefore conducted using the overall $\mu$-$\Lambda$ relation.

### 4.2 Sensitivity of $\mu$-$\Lambda$ relationship to gamma hypothesis

One crucial factor that could affect the $\mu$-$\Lambda$ relationship is the gamma DSD assumption. To investigate this issue, we temporarily

added back all DSDs that were excluded from the previous analysis because they were not conforming to the gamma model, according to the criteria set by Gatidis et al. (2020). For each event, we re-calculated the individual $\mu$-$\Lambda$ relationship and compared the new results to the ones obtained using only the DSDs that satisfied the gamma assumption. In six out of seven cases, the inclusion of the non-gamma cases resulted in larger $\alpha$ and smaller $\beta$ values. However, these changes were not reflected visually in the $\mu$-$\Lambda$ scatterplot as the two opposite changes compensate for each other. Therefore, apart from slightly

changing the parameter values, the gamma hypothesis does not appear to have a strong effect on the overall $\mu$-$\Lambda$ relation. Also, the changes to $\alpha$ (0.518 from 0.514) and $\beta$ (1.328 from 1.339) were rather small and not statistically significant. The fact that the overall $\mu$-$\Lambda$ relation is rather stable with respect to the gamma DSD hypothesis is an interesting result, especially given the fact that there are large differences in sample sizes between non-gamma (1829) and gamma DSDs (652).

### 4.3 Sensitivity of $\mu$-$\Lambda$ relationship to $N_T$

Using the overall relationship from the previous section as a reference, the influence of the number concentration on the $\mu$-$\Lambda$ relationship was investigated. Three different $N_T$ thresholds corresponding to different percentiles of $N_T$ (25%, 50% and 75%) were applied, and only the DSDs with number concentrations above these thresholds were considered. In Fig. 4, the three derived $\mu$-$\Lambda$ relations obtained after applying the $N_T$ filters are shown against the overall relation (no filter). As the $N_T$ threshold is increased from 225 to 300 and 390 m$^{-3}$ (Figs. 4b-d), the $\mu$-$\Lambda$ relation remains relatively stable for lower $\mu$ values,

gradually getting closer to the one proposed by Zhang et al. (2003), especially for higher values of the shape parameter ($\mu$>7). This can be explained partly by the fact that on average, higher $N_T$ values correspond to higher rainfall intensities and larger drop diameters. Also, the average mass-weighted mean diameter increases by approximately 10% as we increase the threshold on $N_T$. This may not represent a big change, but can be enough to slightly affect the $\mu$-$\Lambda$ relation. However, we believe the main reason the $\mu$-$\Lambda$ relation changes with increasing $N_T$ is sampling uncertainty. Indeed, our dataset predominantly features

stratiform rain events with low rainfall intensities, low number concentrations and relatively low and constant mass-weighted mean diameters (see Table 1). As we apply higher thresholds on $N_T$, the DSD samples that only contain a small number of drops and are associated with a higher sampling uncertainty get removed. Consequently, the remaining DSDs with higher number concentrations tend to be associated with lower sampling uncertainties which leads to more reliable $\mu$-$\Lambda$ estimates. Moreover, it is worth pointing out that because of the way $\mu$ is estimated through the cost function in Eq.6, the error distribution

of $\mu$ tends to be positively skewed. On average, we are therefore more likely to overestimate $\mu$ and underestimate the spread of





the DSD rather than the opposite. Since $\mu$ and $\Lambda$ values are positively correlated through their relation with $D_m$ in Eq.5, any overestimated $\mu$ value automatically results in an overestimated $\Lambda$ value (to compensate and get the correct $D_m$). Consequently, as we increase the $N_T$ threshold, sampling errors get reduced and the positively skewed outliers with high $\mu$ and $\Lambda$ values progressively disappear. This removes more and more points on the upper side of the $\mu$-$\Lambda$ curve, pushing the new relation

down towards the one proposed by Zhang et al. (2003). Regarding the sensitivity of the $\alpha$ and $\beta$ parameters describing the $\mu$-$\Lambda$ relationship, our analyses show that they exhibit an opposite behaviour, increasing and decreasing respectively as we increase the threshold on $N_T$. The latter can be attributed to a gradual flattening of the relationship and increase of the intercept parameter. Note that another similar approach to reduce the uncertainty in the estimated $\mu$-$\Lambda$ relationship without applying a threshold on $N_T$ could be to consider longer temporal aggregation intervals than 30 s. However, this would significantly reduce

the amount of data available for analysis.

### 4.4 Influence of sampling resolution on the overall $\mu$-$\Lambda$ relation

In the following, the DSD data corresponding to the seven selected events were re-sampled at four different temporal resolutions of 30, 60, 240 and 480 seconds to investigate the sensitivity of the $\mu$-$\Lambda$ relationship to the choice of the temporal resolution. Similarly to before, only the re-sampled DSDs which satisfied the gamma hypothesis were kept for analysis. Fig. 5 shows

that the overall $\mu$-$\Lambda$ relationship remains very stable, regardless of the considered sampling resolution. Table 2 shows more details about the fitted power-law parameters $\alpha$, $\beta$ at each resolution, including their percentage relative differences against the overall relation at 30 seconds. We can see that the relative error affecting the parameters slightly increases as the temporal resolution is reduced. The latter can be attributed to the lower number of samples available for fitting the parameters. Apart from these obvious sampling effects, the choice of the temporal aggregation scale seems to have very little effect on the overall

$\mu$-$\Lambda$ relationship which remains rather stable across multiple aggregation time scales.

   Note that as we decrease the temporal resolution, the mean values of $\mu$ and $\Lambda$ (Fig. 5) also decrease. This means that there is a progressive transition from peaked DSDs at higher sampling resolutions to broader, more widespread DSDs at lower resolutions. Decreasing the sampling resolution therefore causes the $\mu$ and $\Lambda$ values to shift toward the bottom left part of the scatter plot. However, while the points shift, they remain remarkably close to the initial $\mu$-$\Lambda$ curve derived at the highest

temporal resolution of 30 seconds. The fact that the $\mu$ and $\Lambda$ values change with resolution but that the overall relation between them is preserved across scales suggests that there is a fundamental physical link between certain moments of the DSD, such as the spread and the mean. Also, this relation seems to be quite robust regardless of whether the gamma assumption is valid or not and is only slightly affected by $N_T$. In steady rainfall conditions, it should therefore be possible to use the same $\mu$-$\Lambda$ relationship for DSD retrievals across multiple temporal scales. This is of high importance given the fact that $\mu$-$\Lambda$ relations

are often used to retrieve DSDs from radar observations, which have different sampling volumes and levels of aggregation than disdrometer data. Moreover, the use of a $\mu$-$\Lambda$ relationship may still be justified from a physical point of view, even if the underlying DSDs do not strictly comply with the gamma distribution hypothesis. Obviously, the fact that we have selected relatively similar, stratiform events with low rainfall intensities and low temporal variability is a crucial factor here since it means that by resampling, we do not significantly change the properties of the DSDs or mix together different rainfall regimes.





By contrast, larger differences in $\mu$-$\Lambda$ relationships can be expected for mixed-type rainfall events with multiple and rapid alternations between stratiform and convective rain.

     On the other hand there is still substantial controversy in the literature around the reason why $\mu$-$\Lambda$ relations exist in the first place and why certain DSD parameters are linked to each other. One justification could be that the effective number of parameters needed to describe most DSDs is probably less than three. In other words, under proper normalization, all DSDs look

rather similar to each other. For example, Torres et al. (1994) introduced a single DSD normalization technique based on one reference moment (usually the rain rate). Later, Testud et al. (2001); Lee et al. (2004) proposed a more general normalization technique based on two reference moments (usually the $3^{rd}$ and $6^{th}$ moments). The existence of a $\mu$-$\Lambda$ relationship may just be the consequence of such scaling laws. In their study, Moisseev and Chandrasekar (2007) have also argued that data filtering can have a strong influence on the relation itself, leading to spurious links between $\mu$ and $\Lambda$. However, this is not the case in our

study. On the contrary, our results show that when events with similar characteristics are chosen, the overall $\mu$-$\Lambda$ relationship can be rather stable, barely depending on the different filters applied to the data (e.g. inclusion/exclusion of non gamma DSDs or minimum threshold for $Z_{hh}$ and $Z_{dr}$). Other studies have pointed out that the constraints linking $\mu$ and $\Lambda$ during parameter fitting can lead to correlated errors between estimated gamma DSD parameters and biased relationships (Williams et al., 2014; Moisseev and Chandrasekar, 2007). Indeed, because of the way we fit $\mu$ and $\Lambda$ through $D_m$ (see Section 3a, DSD model), the

parameters end up positively correlated with each other. In other words, if $\mu$ is overestimated, $\Lambda$ will also be overestimated because it has to compensate for the bias in $\mu$. To address this, Williams et al. (2014) proposed a $\sigma$'-$D_m$ relationship, where $D_m$ is the mass-weighted mean diameter and $\sigma$' a new mass spectrum standard deviation, defined and constructed to be statistically independent of $D_m$. Even though their approach seems to lead to smaller biases, our results show that it is also possible to derive reliable $\mu$-$\Lambda$ relationships without defining a new $\sigma$, simply by excluding the non-gamma DSDs cases and carefully

filtering out DSDs with very low $N_T$ values.

## 5   Sensitivity of DSD retrievals

In this section, the sensitivity of the DSD retrieval method as a whole is evaluated. First, the TARA and Parsivel observations are compared with each other to highlight their differences and understand how possible biases in reflectivity or differential reflectivity affect the accuracy of the retrievals. Then, the sensitivity of the retrieved DSD parameters to different bias corrections,

scale corrections and data filters is quantified and possible ways to mitigate errors during retrievals are proposed.

### 5.1   Overall agreement between radar and disdrometer

### 5.1.1   Agreement of $Z_{hh}$ and $Z_{dr}$ observations between TARA and Parsivel

In this section the agreement between the Parsivel and TARA measurements is investigated. The goal is to quantify how well the measurements of the two sensors agree with each other before the DSD retrievals. Fig. 6 shows the scatter plots of the

reflectivity factor ($Z_{hh}$, top) and differential reflectivity ($Z_{dr}$, bottom) from the disdrometer versus TARA at 200 m height.





For this first comparison, the $Z_{hh}$ and $Z_{vv}$ measurements of TARA were aggregated (in linear scale) to 30 seconds in order to be comparable with the disdrometer data. No other additional filter was applied. Fig. 6a shows that $Z_{hh}$ measurements are highly correlated (correlation coefficient=0.94). However, the radar significantly underestimates $Z_{hh}$ compared with the disdrometer. The offset in $Z_{hh}$ slightly varies with time but is in the order of 6 to 7 dBZ (overall bias 6.44 dBZ). Additional

bias analyses at a different height of 400 m show that the offset does not change substantially with height, which suggests that the FMCW incomplete beam overlap correction at near ranges (see Section 2b, radar data) works well and that the offset in reflectivity is likely due to calibration issues of TARA rather than range-related issues. Unlike $Z_{hh}$, the differential reflectivity ($Z_{dr}$) measurements appear to be in much better agreement with the disdrometer (overall bias -0.03dB), as can be seen in the bottom panel of Fig. 6. However, the correlation for $Z_{dr}$ is lower (correlation coefficient=0.71) and there is significant scatter,

especially for higher values of $Z_{dr}$. Note that the vast majority of $Z_{dr}$ values are small (less than 0.2 dB), which makes sense given that we are mostly dealing with light stratiform rain and that the elevation angle of 45 degrees in TARA further reduces the magnitude of $Z_{dr}$.

### 5.1.2 $Z_{hh}$-$Z_{dr}$ relationships for TARA and Parsivel

In the top panel of Fig. 7, the $Z_{hh}$-$Z_{dr}$ relation of each sensor is presented. It shows that most of the time, TARA measures

higher $Z_{dr}$ values for a given $Z_{hh}$ than the disdrometer. Once the calibration bias in $Z_{hh}$ is removed (Fig. 7, bottom), the agreement improves and the radar and disdrometer-derived relationships nicely overlap with each other. Nevertheless, and despite the bias correction, TARA still tends to measure slightly higher $Z_{dr}$ values than the Parsivel for a given $Z_{hh}$. This can be due to a difference in height or scale between the two measurements. The absence of a clear relation between $Z_{hh}$ and $Z_{dr}$ is not really a problem for the DSD retrieval method itself. In fact, a relation between $Z_{hh}$ and $Z_{dr}$ is not always expected

since $Z_{hh}$ does depend on $N_T$ while $Z_{dr}$ does not. However the fact that TARA and the Parsivel disdrometer exhibit different $Z_{hh}$-$Z_{dr}$ relationships might negatively impact the accuracy and consistency of the retrieved DSDs.

### 5.1.3 First retrievals

In the following, we apply the DSD retrieval method described in Section 3c using $Z_{hh}$ and $Z_{dr}$ measurements from TARA and compare the results to the disdrometer data at 30 seconds resolution. For the retrievals, we used the overall $\mu$-$\Lambda$ relationship

inferred in Section 3c (DSD retrieval method), from the disdrometer observations at 30 seconds sampling resolution.

For illustration purposes, the event on 11 October 2014 was chosen. The time series of retrieved $\mu$, $D_m$, $N_T$ and observed $Z_{hh}$ and $Z_{dr}$ values for this event are presented in Fig. 8 and Fig. 9, top. Overall, we see that there is a rather good agreement in terms of the retrieved $\mu$ and $D_m$ values as long as the $Z_{dr}$ values are not too low (i.e., >0.1 dB). When $Z_{dr}$ is low (e.g., between 12:20-13:15 UTC), we see that the retrievals become very uncertain, exhibiting much larger fluctuations over time.

Compared with $\mu$, the retrieved $N_T$ values are substantially more uncertain. There are some outliers and, on average, the retrieved $N_T$ values from TARA are about 100 m$^{-3}$ lower than those from the Parsivel disdrometer. This bias is attributed to the 6-7 dB offset in $Z_{hh}$ in TARA which propagates non-linearly to $N_T$ through the link between $Z_{hh}$ and $N_T$ in Eqs. 9-10. On the other hand, we also see some isolated cases where $N_T$ is overestimated, such as at the beginning (10:57 UTC) and the end





(13:15 and 13:23 UTC) of the event. These periods are characterized by underestimated $Z_{dr}$ and $D_m$ values by TARA which, in combination with the relatively high $Z_{hh}$ values, leads to an overestimation of $N_T$.

For a better overview, the retrieved DSD parameters ($\mu$, $D_m$ and $N_T$) for all selected events are plotted against the ones from the disdrometer in Fig. 10. We can see that the retrieved $\mu$ values from the radar tend to be lower compared with the disdrometer. The overall bias on the retrieved $\mu$ values is 2.11, which is rather large and is not immediately apparent from the case study on October $11^{th}$ (Fig. 8). Note that the retrieved $\mu$ values from TARA can never exceed 8 due to the 0.1 dB cutoff applied to $Z_{dr}$ observations (very light rain, peaked DSDs). Because of this, there is a slight conditional bias on the retrieved $\mu$ values for low $Z_{dr}$ values. Since $\mu$ values are unaffected by the bias in reflectivity and $Z_{dr}$ measurements appear to be well calibrated, the bias we see in $\mu$ values must either be due to the $\mu$-$\Lambda$ relationship or to differences in scale, height and measurement principles between the two sensors. Unlike $\mu$, there is better agreement for $D_m$ retrievals with -0.09 overall bias. This is the case for the case study on October $11^{th}$ as well, where $D_m$ retrievals from Parsivel and TARA are almost similar throughout the event (Fig. 8, middle) except for the period between 12:45 and 13:00 when $Z_{dr}$ is low. Looking at the number concentration (Fig. 10, bottom), we see a significant underestimation in $N_T$ from TARA (overall bias=276 m$^{-3}$, multiplicative bias=4.52), which can be explained by the large 6.44 dBZ bias on $Z_{hh}$ in TARA and is consistent with the previously reported underestimation for the event on 11 October 2014.

Despite the fact that $N_T$ values tend to be underestimated on average, we can also see several large spikes in retrieved $N_T$ values, such as during the second half of the case study event (Fig. 9, top). If we perform a more in-depth analysis of this period (i.e., between 12:30 and 13:30 UTC) in Fig. 9, middle and compare it with the $Z_{hh}$ and $Z_{dr}$ observations of the corresponding period (Fig. 9, bottom), we see that all five spikes in $N_T$ correspond to low values of $Z_{dr}$ and relatively high $Z_{hh}$ values. The low $Z_{dr}$ leads to large $\mu$ values and underestimated raindrop sizes during the retrieval. To compensate for this and achieve the correct reflectivity, $N_T$ needs to be increased by a lot. Note that spikes in $N_T$ can still occur even if $Z_{hh}$ is modest or decreasing locally, as long as $Z_{dr}$ is very small, as for example for the spikes No 2 and 3 there is a local maximum for $Z_{hh}$ while for the other spikes the $Z_{hh}$ decreases.

The differences documented above are important because they show that DSD retrievals can be very sensitive to combined biases in $Z_{dr}$ and $Z_{hh}$ relative to each other. The latter can be linked to calibration issues. However, inconsistencies can also arise due to differences in height, sampling volumes and temporal aggregation scales between radar and disdrometer measurements.

## 5.2 Sensitivity to calibration bias correction

Given the systematic underestimation of the reflectivity factor in TARA, a bias correction was applied before proceeding with the DSD retrievals. Indeed, the bias correction was considered essential to get more reliable results, especially for $N_T$. Since the $N_T$ retrievals require the reflectivity to be converted from logarithmic (dB) to linear scale (mm$^6$m$^{-3}$), a multiplicative adjustment factor known as the G/R ratio (i.e., the ratio of the sum of Parsivel to TARA reflectivity values) was used to bias-correct the TARA measurements, treating the disdrometer observations as the reference truth. The value of the G/R ratio was 4.52, which confirmed the large calibration bias of TARA. To address the bias, all TARA reflectivity values (in linear scale)





were multiplied by 4.52 and the new DSD parameters were retrieved. As expected, the first two DSD parameters $\mu$ and $D_m$, were completely unaffected by the bias adjustment, as they only depend on $Z_{dr}$ (see Section 3c, DSD retrieval method). Fig. 11

on the other hand, shows that $N_T$ retrievals were substantially improved, and the bias decreased from 276 m$^{-3}$ to 89 m$^{-3}$. Despite the lower bias, we can see that large uncertainties remain in the retrieved $N_T$ values, as highlighted by the large scatter and frequent outliers.

### 5.3    Sensitivity to scale bias correction

In the following, a small, additional, bias adjustment was applied to $Z_{dr}$ to try to account for the large difference in sampling

volumes between the TARA radar and the Parsivel disdrometer. This second adjustment is conceptually different from the one applied to $Z_{hh}$, which was primarily due to calibration issues. Contrarily to $Z_{hh}$, the differential reflectivity $Z_{dr}$ of TARA is assumed to be well-calibrated. Therefore, the differences in mean and standard deviation are primarily attributed to differences in scale, height and measurement principles. Note that this scale bias also applies to $Z_{hh}$. However, for $Z_{hh}$, the effect is masked by the large calibration bias and the two cannot be separated.

According to Fig. 6, bottom, the average $Z_{dr}$ values measured by TARA are 0.03 dB larger than the ones from the Parsivel disdrometer, which makes sense given that the radar sees a larger measurement volume, which makes it more likely to contain at least a few larger drops. Even though a 0.03 dB difference seems small, such a bias can have a significant effect on the DSD retrievals given that the majority of $Z_{dr}$ values are rather small (e.g., between 0.1 and 0.2 dB). A 0.03 dB bias on $Z_{dr}$ therefore represents a relative error of 15-30%.

Fig. 12 shows the retrieved DSD parameters after correcting for the scale bias. We see a reduction of the bias affecting $\mu$ and $D_m$, which are directly linked to $Z_{dr}$. The bias affecting $\mu$ is halved from 2.11 to 1.12 and the bias affecting $D_m$ is reduced from -0.09 mm to -0.02 mm. The correlation coefficient remains relatively stable, regardless of the scale correction. Despite the improvements for $\mu$ and $D_m$, the $N_T$ retrievals remain problematic, with low correlation coefficient of 0.12 (compared to 0.17 without scale bias correction) and moderate bias of -32 m$^{-3}$ (compared to 89 m$^{-3}$ without correction). Also, the average

$N_T$ value increased significantly, from 261 m$^{-3}$ to 382 m$^{-3}$ (+46%) which highlights the large sensitivity of $N_T$ to changes in the differential reflectivity.

### 5.4    Sensitivity of $N_T$ to outliers

The results presented in the previous sections have shown that unlike $\mu$ and $D_m$, the uncertainty surrounding the $N_T$ retrievals tend to be much larger. This can be explained by the fact that $N_T$ is the last parameter to be retrieved in Eq. 10, which makes

it more susceptible to error propagation/accumulation during the first steps of the retrieval procedure. Errors on retrieved $N_T$ values can be due to the retrieval method itself (e.g., the assumed $\mu$-$\Lambda$ relation and gamma DSD model), biased radar observations (e.g., calibration errors in $Z_{hh}$ or/and $Z_{dr}$) or additional biases due to differences in measurement scale, height and principle between radar and disdrometers. Considering the fact that the events used in this study mainly consist of weak/light stratiform rain, the errors/uncertainty affecting the measured $Z_{dr}$ values are very likely to play an important role.



The scatterplot of retrieved $N_T$ values versus disdrometer data in Fig. 10, bottom, shows a low correlation coefficient and a significant underestimation from TARA mainly due to the huge bias in $Z_{hh}$ (6.44 dBZ). However, it is worth noticing that even after applying a calibration bias correction on $Z_{hh}$, there was no substantial improvement in terms of the $N_T$ retrievals (Fig. 11). Even though the bias in $N_T$ was reduced (89 m$^{-3}$ compared to 276 m$^{-3}$), the scatter increased and the correlation coefficient remained low (0.17). The scale correction for $Z_{dr}$ results in an even worse agreement (correlation coefficient 0.12,

Fig. 12 bottom). In general, two distinct groups of data points with drastically different error properties can be seen. For the first, the retrieved $N_T$ values are severely overestimated compared to the Parsivel disdrometer, by up to one order of magnitude. For the second group, the retrieved $N_T$ values are up to ten times lower than the disdrometer values.

The conclusion is that there are two different types of combinations of $Z_{hh}/Z_{dr}$ that result in unreliable $N_T$ retrievals. The first group is comprised of low $Z_{dr}$ values compared to $Z_{hh}$, which results in overestimated $N_T$ values. These are all the pairs

of $Z_{hh}/Z_{dr}$ in the lower right part of Fig. 14. Since $Z_{dr}$ is low, the only way to get a high reflectivity is by increasing $N_T$. The second group consists of relatively high $Z_{dr}$ values compared to $Z_{hh}$, which leads to underestimated $N_T$ values. These points correspond to the top left part of Fig. 14. Since $Z_{dr}$ is large, the only way to get a low $Z_{hh}$ is to decrease $N_T$. Together, these two different types of outliers are responsible for the large scatter observed in retrieved $N_T$ values.

Each retrieval has its own uncertainty and error characteristic, depending on the pair of $Z_{hh}/Z_{dr}$. For example, the scale

correction has different impacts on the different subgroups. Even though there is a general increase in $N_T$ to compensate for the new reduced value of $Z_{dr}$, the aforementioned correction had a significant impact on the subgroup which corresponds to the points which are overestimated by TARA and negligible for the ones which are underestimated.

A possible way to reduce the uncertainty affecting the $N_T$ retrievals and thereby avoiding large errors is to filter out all potentially problematic combinations of $Z_{hh}/Z_{dr}$. In the following, a filter which aims at controlling the uncertainty on $N_T$

by removing certain $Z_{hh}/Z_{dr}$ combinations which are difficult to handle is applied. Note that these "outliers" in the $Z_{hh}/Z_{dr}$ space are not necessarily wrong. They are just problematic in the sense that they can potentially result in very large errors in terms of retrieved $N_T$. The applied filter is two-dimensional depending on both $Z_{hh}$ and $Z_{dr}$ values since the uncertainty derives from their combination. A power-law model was used to fit the radar observables $Z_{hh}$ and $Z_{dr}$ after calibration and scale bias correction, respectively. Based on that model, an upper and lower curve defining the limits of acceptable $Z_{hh}$ and

$Z_{dr}$ pairs is obtained by adding respectively subtracting a given tolerance from $Z_{hh}$ as in Fig. 14. For illustration purposes $\pm$ 6 dB was selected, but several other options (i.e., 2, 4 and 8 dB) were examined as well. Table 3 lists all options together with their corresponding performances for $\mu$, $D_m$ and $N_T$. We see that by removing certain points beyond the lower and upper limits in the $Z_{hh}/Z_{dr}$ space, it is possible to improve the correlation between the observed and retrieved $\mu$, $D_m$ and $N_T$ values while keeping a similar bias. For $\mu$ and $D_m$, the best tolerance (in terms of correlation) seems to be $\pm$ 2 dB and $\pm$ 4 dB.

However, these are rather strict, which means that a large fraction of the data points would have to be discarded (i.e, 56% and 23% respectively) for a modest gain in performance. For the $N_T$ retrievals, the optimal tolerance appears to be $\pm$ 6 dB, which discards less than 9% of the data but still manages to significantly increase the correlation (0.12 to 0.24) and decrease the absolute value of the bias (-32 to 10 m$^{-3}$). Note that contrarily to $\mu$ and $D_m$, filtering out more data points does not necessarily





increase the performance in terms of the $N_T$ retrievals. Fig. 13 shows the final radar DSD retrieval results, after applying a
filter with a tolerance of $\pm\ 6$ dB.

## 6    Conclusions

A previously proposed method for retrieving DSDs based on radar reflectivity measurements ($Z_{hh}$), differential reflectivity
($Z_{dr}$) and an empirical relation between the shape ($\mu$) and slope ($\Lambda$) parameters of a gamma DSD model was investigated.
Observations from a nearby optical disdrometer were used to derive the $\mu$-$\Lambda$ relationship as well as for performing an indi-
rect validation of the retrieved DSDs. While the retrieval method itself is well-known, this study primarily focused on the
critical assumptions behind it, in order to outline potential sources of errors and uncertainties. First, a thorough sensitivity
analysis of the $\mu$-$\Lambda$ relation to various factors such as the temporal sampling resolution, the adequacy of the gamma model
hypothesis, sensitivity to the concentration number ($N_T$) and event by event variations was conducted. Then, the influence of
calibration errors in radar observations, and scale differences between radar and disdrometer observations were highlighted
and investigated. Finally, a filter designed to mitigate uncertainty during $N_T$ retrievals was proposed. According to the results
the following conclusions can be drawn.

1. The $\mu$-$\Lambda$ relationship derived from a nearby disdrometer proved quite robust to the choice of the temporal sampling
   resolution, validity of the gamma model hypothesis, sample size and event by event variability. However, only seven,
   rather similar stratiform rain events were considered. More research is necessary to fully understand and quantify inter-
465       event variability of $\mu$-$\Lambda$ relationships in convective rain.

2. Radar calibration biases significantly affect the accuracy and reliability of the retrieved DSDs. Both $Z_{hh}$ and $Z_{dr}$ must
   be bias-corrected before retrieving the DSD.

3. Even for well-calibrated radars, a small, additional bias correction to account for the scale difference between radar and
   disdrometer observations can be useful to reduce conditional biases in retrieved μ and $N_T$ values.

4. Finding the right bias and scale corrections for $Z_{hh}$ and $Z_{dr}$ is not straightforward. Often the bias due to scale differences
   cannot be separated from the bias due to calibration errors and measurement noise. In our case, $Z_{dr}$ was very well
   calibrated which allowed us to investigate the scale correction in more detail. However, due to the large calibration
   offset, the scale correction for $Z_{hh}$ could not be determined.

5. Despite our best efforts, the retrieved $N_T$ values remained highly uncertain. Two different types of outliers were iden-
475       tified, resulting in severely underestimated or overestimated $N_T$ values. A simple filter for removing outliers in the
   $Z_{hh}$/$Z_{dr}$ space was proposed. The filter gets rid of some problematic cases, which slightly improves the reliability of
   the $N_T$ retrievals. But improvements remained modest and removing more data did not systematically result in better
   performances.





Finally, it should be mentioned that we do not expect the exact same adjustments to hold for other DSD retrieval algorithms
or radar systems. The adjustments mentioned in this study are specific to the TARA radar and Parsivel optical disdrometer.
For example, the radar elevation angle was 45 degrees, which is not ideal for such retrievals. Uncertainties for lower elevation
angles would probably be smaller due to higher $Z_{dr}$ values. Depending on the radar system, more elaborate corrections than a
simple shift in $Z_{dr}$ might be necessary to achieve optimal performance across a larger number of rain events. Similarly, more
convective rain events should be included to study the performance and reliability of DSD retrievals based on $\mu$-$\Lambda$ relationships
during heavy convective rain with larger drop sizes. Finally, future work could look at the importance of $\mu$-$\Lambda$ relations in DSD
retrievals from other relevant rainfall sensors, such as satellite observations, which have much larger sampling volumes and
errors than ground-based radar and for which the scale corrections might therefore play a more important role.



*Data availability.* The DSD data, collected by Parsivel optical disdrometer in the Netherlands during the ACCEPT campaign and used in this study are available.

*Author contributions.* Christos Gatidis mainly worked on data processing, visualization of the results and writing – original draft preparation. Marc Schleiss and Christine Unal focused on the supervision of Christos with fundamental ideas about the direction of the research, the used methodology and finally the writing – review and editing

*Competing interests.* The authors declare that they have no conflict of interest.

*Acknowledgements.* This work was supported by the Netherlands Organisation for scientific research (NWO) through the "User Support
Programme Space Research 2012-2016", project ALW-GO/15-35.





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



**Table 1.** Overview of the selected events. Date, duration, number of samples, average rain intensity ($\overline{RR}$), average mass-weighted mean diameter ($\overline{D_m}$), average number concentration ($\overline{N_T}$), parameters of the $\mu$-$\Lambda$ relationship ($\alpha$, $\beta$) and their corresponding percentage relative errors. Note that only the DSDs conforming to the gamma model (see Section 3a, DSD model) were considered when computing these statistics.

| Event | Date | Duration (hh:mm) | No. of samples | $\overline{RR}$ (mm h$^{-1}$) | $\overline{D_m}$ (mm) | $\overline{N_T}$ (m$^{-3}$) | $\alpha$ | Percentage relative error $\alpha$ (%) | $\beta$ | Percentage relative error $\beta$ (%) |
|---|---|---|---|---|---|---|---|---|---|---|
| 1 | 8 Oct | 2:00 | 77 | 1.22 | 1.08 | 279 | 0.514 | 0.00 | 1.347 | 0.59 |
| 2 | 11 Oct | 3:15 | 88 | 1.81 | 1.12 | 383 | 0.227 | -126.76 | 1.720 | 22.12 |
| 3 | 15 Oct | 2:30 | 147 | 0.86 | 0.9 | 295 | 0.676 | 24.04 | 1.241 | -7.92 |
| 4 | 16 Oct | 2:20 | 110 | 2.46 | 1.18 | 418 | 0.354 | -45.32 | 1.494 | 10.36 |
| 5 | 24 Oct A' | 2:00 | 38 | 1.0 | 1.02 | 254 | 0.415 | -23.73 | 1.410 | 5.01 |
| 6 | 24 Oct B' | 2:00 | 27 | 2.76 | 1.44 | 315 | 0.178 | -187.91 | 1.795 | 25.37 |
| 7 | 3 Nov | 4:25 | 165 | 0.78 | 0.92 | 292 | 0.832 | 38.23 | 1.144 | -17.08 |
| Overall | - | 18:30 | 652 | 1.37 | 1.03 | 323 | 0.514 | - | 1.339 | - |





**Table 2.** The parameters of the $\mu$-$\Lambda$ relationship ($\alpha$, $\beta$) for different sampling resolutions and their percentage relative error against the corresponding values at 30 seconds.

| Resolution (sec) | $\alpha$ | Percentage relative error $\alpha$ (%) | $\beta$ | Percentage relative error $\beta$ (%) | No. of samples |
|---|---|---|---|---|---|
| 30 | 0.514 | - | 1.339 | - | 652 |
| 60 | 0.518 | 0.84 | 1.337 | -0.25 | 519 |
| 240 | 0.529 | 2.75 | 1.329 | -0.83 | 200 |
| 480 | 0.527 | 2.52 | 1.328 | -0.88 | 115 |





**Table 3.** Filter performance (correlation coefficient, bias) of DSD retrievals ($\mu$, $D_m$ and $N_T$) for different levels of tolerance ($\pm$ 2, 4, 6, 8 and 10 dBZ).

| $\pm$ dBZ | % of data removed | $\mu$ (correlation coefficient, bias) | $D_m$ (correlation coefficient, bias) | $N_T$ (correlation coefficient, bias) |
|---|---|---|---|---|
| 10 (No filter) | 0 | 0.57 / 1.12 | 0.74 / -0.02 | 0.12 / -32 |
| 8 | 2.34 | 0.59 / 1.19 | 0.75 / -0.02 | 0.20 / -17 |
| 6 | 8.57 | 0.60 / 1.14 | 0.78 / -0.03 | 0.24 / 10 |
| 4 | 23.12 | 0.61 / 1.06 | 0.81 / -0.03 | 0.21 / 33 |
| 2 | 56.36 | 0.62 / 1.20 | 0.85 / -0.03 | 0.15 / 51 |





**Figure 1.** Height-time plots (top to bottom) of reflectivity factor (dBZ) and differential reflectivity (dB) on 11 October, 2014.

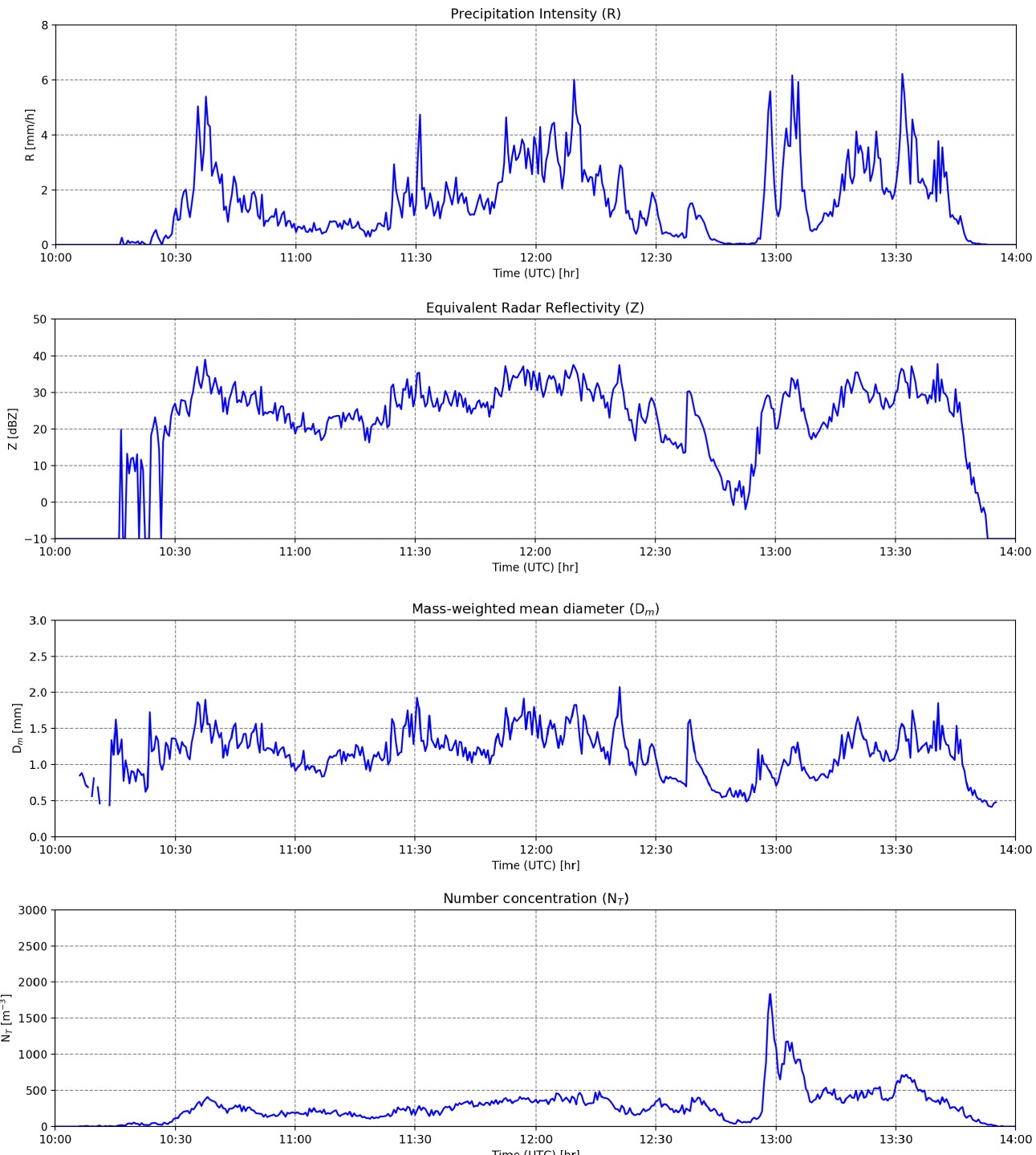

**Figure 2.** Time series of (top to bottom) precipitation intensity [mmh$^{-1}$], reflectivity factor [dBZ], mass-weighted mean diameter [mm] and number concentration [m$^{-3}$] from Parsivel disdrometer data on 11 Oct 2014.

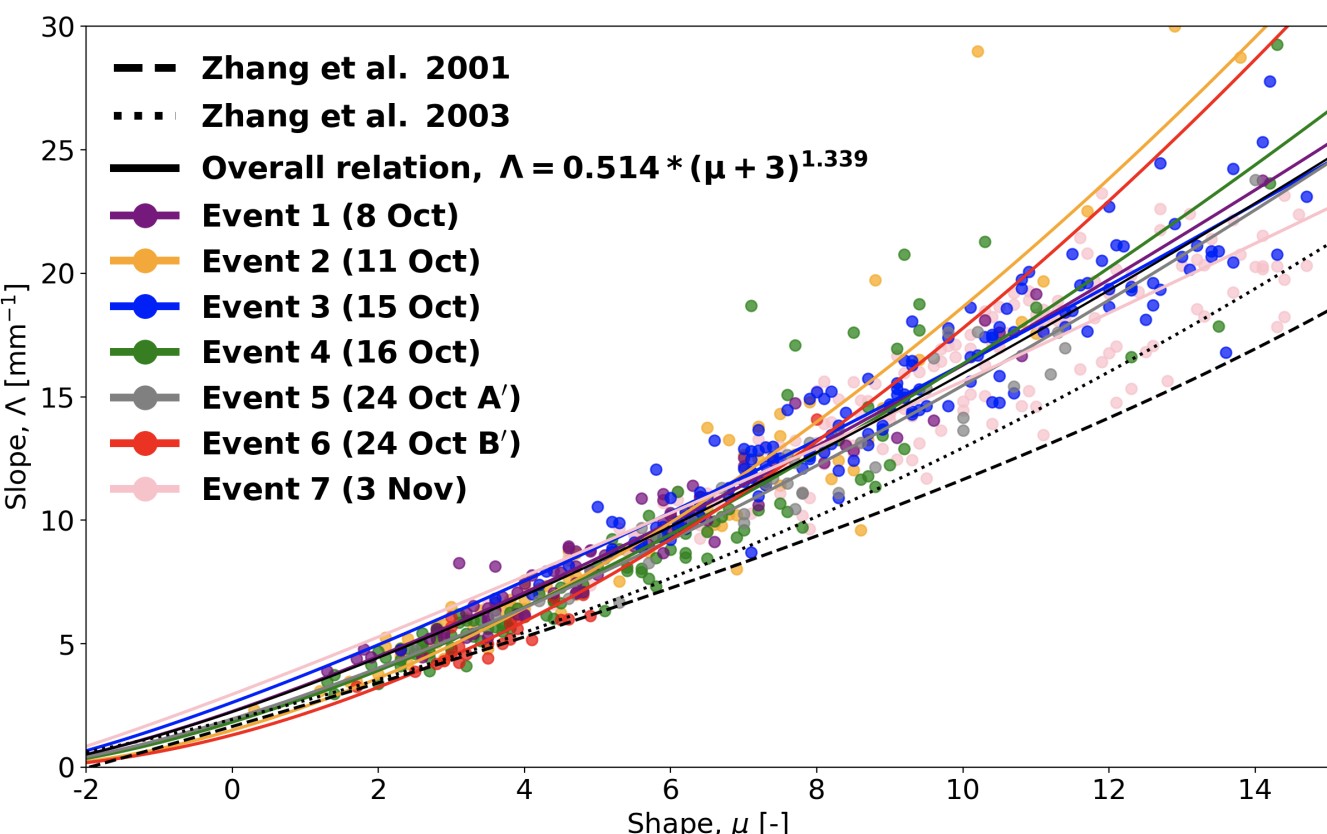

**Figure 3.** Scatter plot between $\mu$ and $\Lambda$ of the selected events colored by event (only gamma DSDs were considered). The $\mu$-$\Lambda$ relationship of each event was fitted and plotted against the overall relationship. The proposed relations by Zhang et al. 2001 and 2003 were plotted as a reference from the literature.

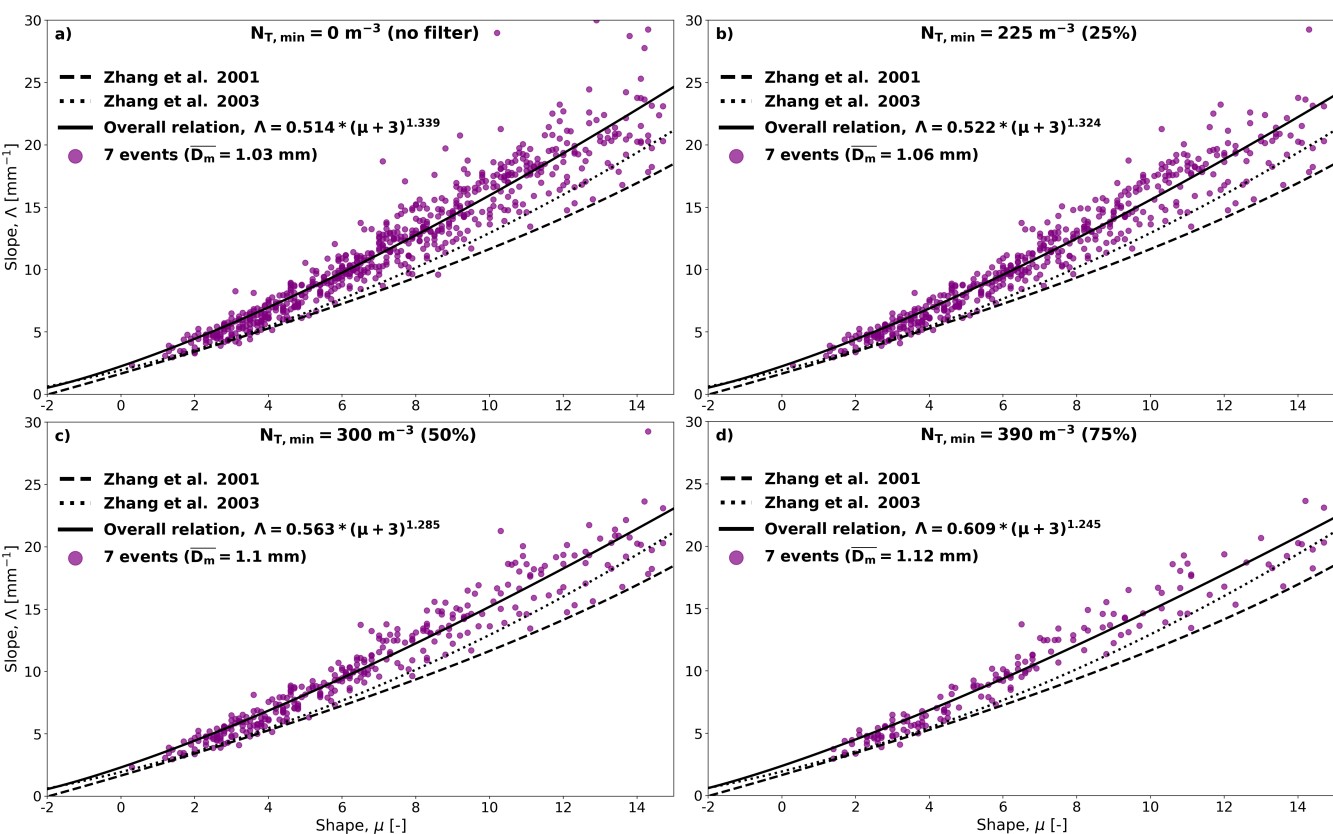

**Figure 4.** Four scatter plots between $\mu$ and $\Lambda$ of the selected events using four different minimum $N_T$ thresholds corresponding to different percentiles of $N_T$. The $\mu$-$\Lambda$ relationship of each $N_T$ threshold was fitted and plotted against the proposed relations by Zhang et al. 2001 and 2003. (a) $N_{T,min}$=0 m$^{-3}$ (no filter), (b) $N_{T,min}$=225 m$^{-3}$, (c) $N_{T,min}$=300 m$^{-3}$ and (d) $N_{T,min}$=390 m$^{-3}$.

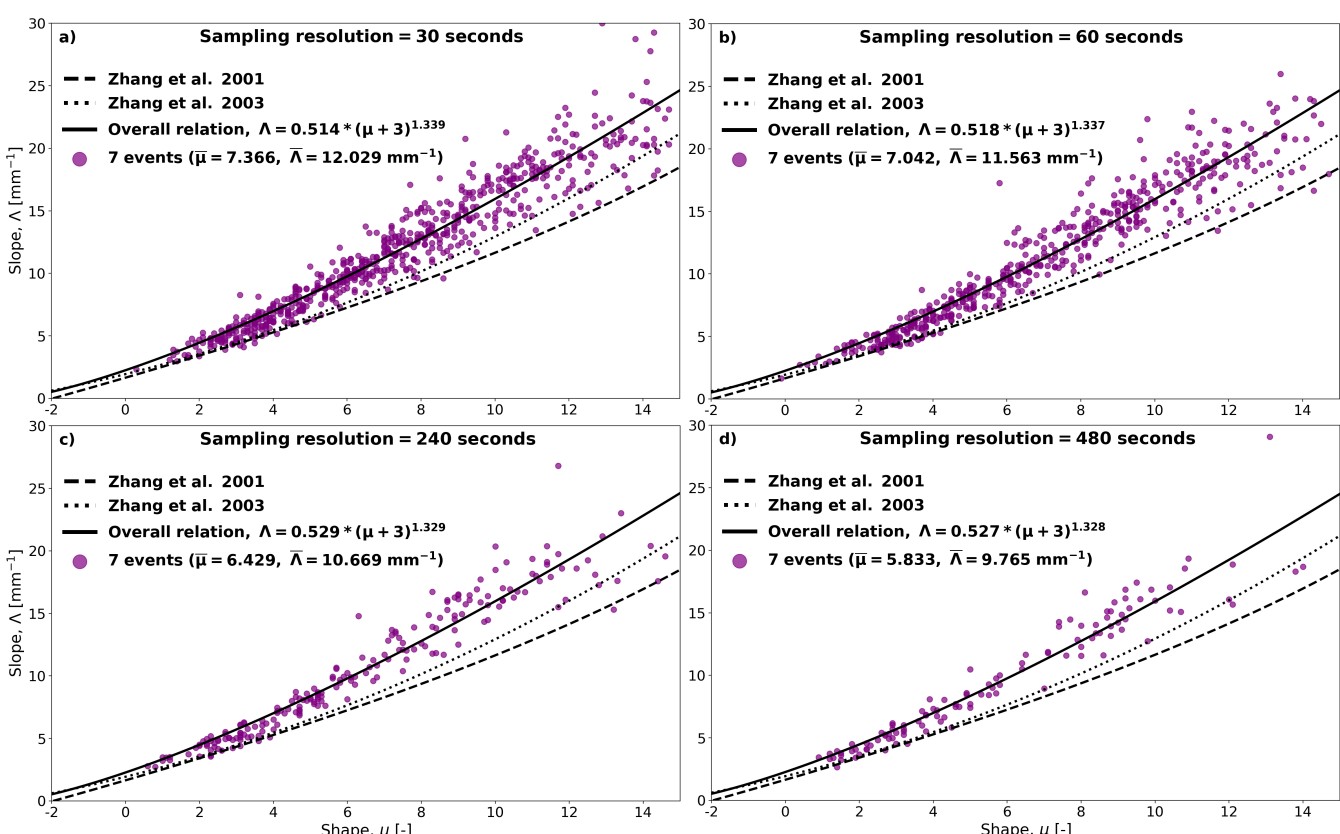

**Figure 5.** Four scatter plots between $\mu$ and $\Lambda$ of the selected events using different resolutions. The $\mu$-$\Lambda$ relationship of each resolution was fitted and plotted against the proposed relations by Zhang et al. 2001 and 2003. (a) 30 s, (b) 60 s, (c) 240 s and (d) 480 s.



**Figure 6.** Scatterplot between the observations of $Z_{hh}$ (dBZ) and $Z_{dr}$ (dB) from the disdrometer and the radar.

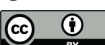



**Figure 7.** $Z_{hh}$-$Z_{dr}$ relations between the disdrometer and the radar (top to bottom) before and after the calibration bias in $Z_{hh}$ is removed.



**Figure 8.** Time series of the DSD retrievals ($\mu$ and $D_m$), and $Z_{hh}$ and $Z_{dr}$ observations from the disdrometer and the radar.

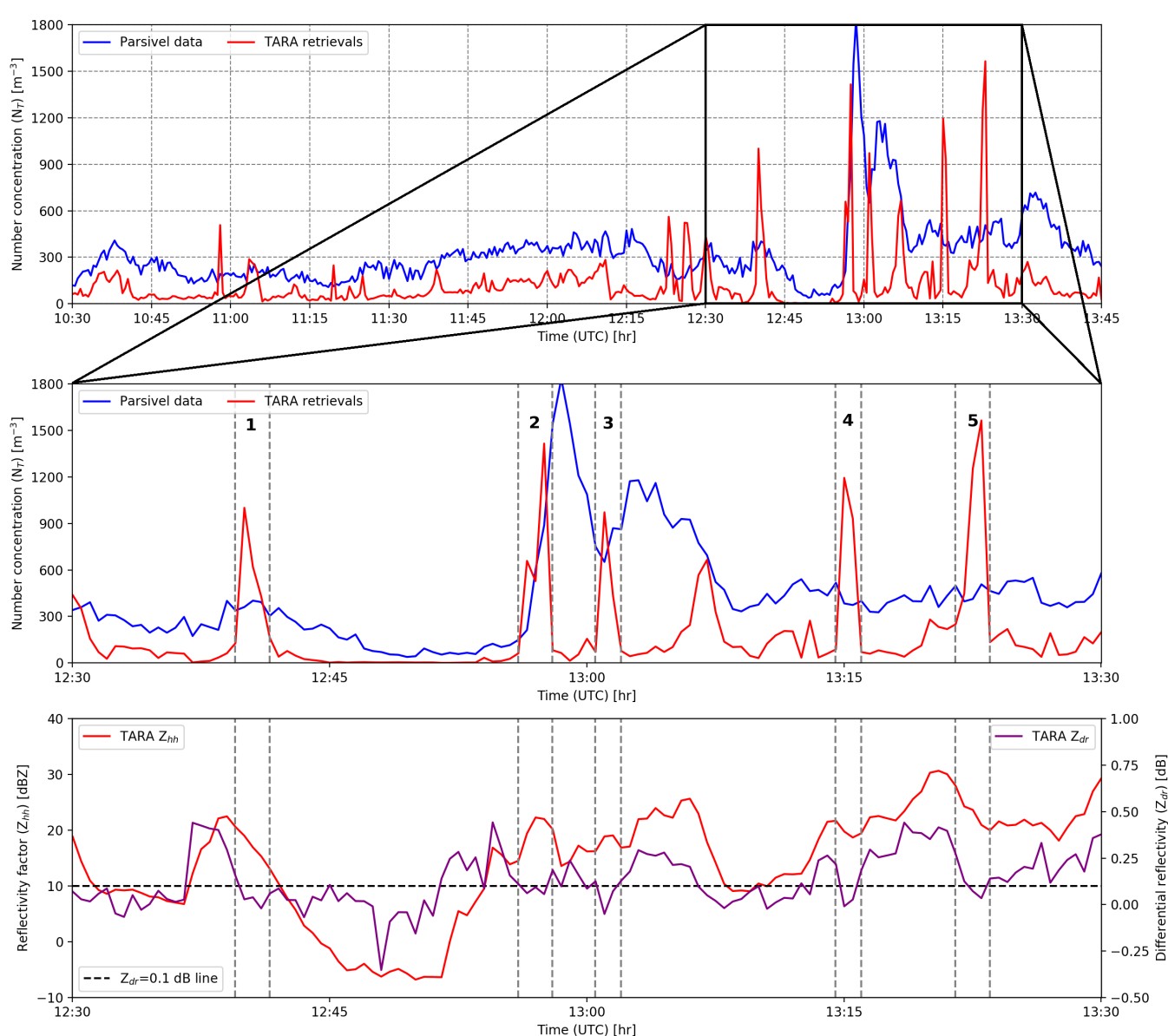

**Figure 9.** Time series of the $N_T$ retrievals (top), zoomed version for the period between 12:30 and 13:30 UTC (middle) and the corresponding $Z_{hh}$ and $Z_{dr}$ observations from the disdrometer and the radar (bottom).







**Figure 10.** Scatterplot of DSD retrievals ($\mu$, $D_m$ and $N_T$) between radar and disdrometer.





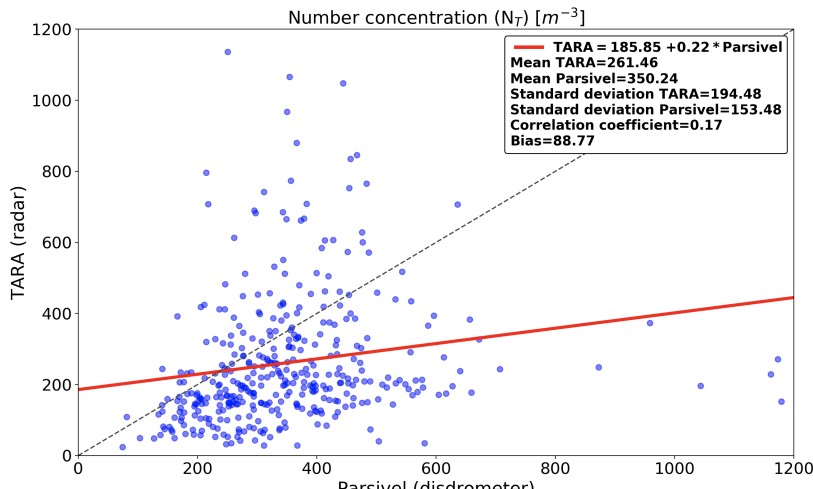

**Figure 11.** Scatterplot of $N_T$ retrievals between radar and disdrometer after applying the calibration bias correction on $Z_{hh}$.





**Figure 12.** Scatterplot of DSD retrievals between radar and disdrometer after applying the scale bias correction on $Z_{dr}$.



**Figure 13.** Scatterplot of DSD retrievals between radar and disdrometer after applying the $Z_{hh}$ - $Z_{dr}$ relation outlier removal.





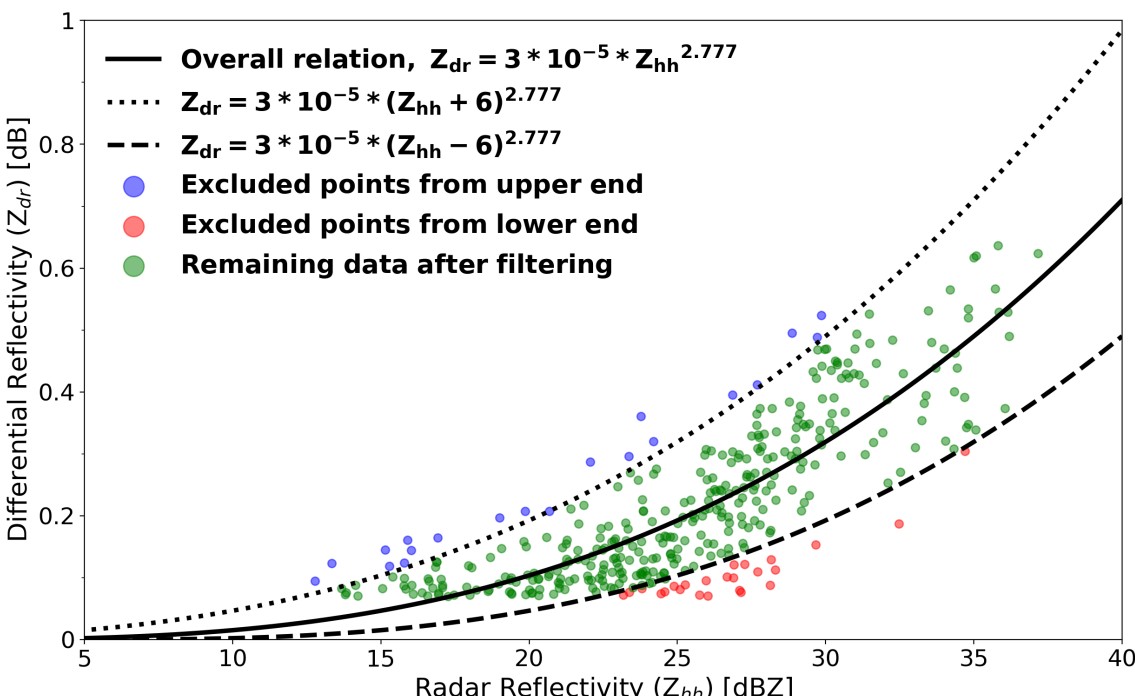

**Figure 14.** Example of the filtering based on the $Z_{hh}$ - $Z_{dr}$ relationship with the overall power-law fit and the corresponding ones for the upper and lower end using $\pm$ 6 dBZ.