# Peer review of "Sensitivity analysis of DSD retrievals from polarimetric radar in stratiform rain based on $\mu$ - $\Lambda$ relationship"

_Atmospheric Measurement Techniques, 2022_

## Referee Comment (RC1)

**REVIEW REPORT**

Review of amt-2022-92

By Christos Gatidis, Marc Schleiss, and Christine Unal

Manuscript Title – Sensitivity analysis of DSD retrievals from polarimetric radar in stratiform rain based on μ-λ relationship

**GENERAL COMMENTS**

The Authors evaluated the effects that different aspects/assumptions can have on the mu-lambda relations and retrieved the mu and lambda parameters from and S-band radar and compared the retrievals with a disdrometer.

The papers il well organized and the methodology and results are well described. I suggest the publication on AMT after addressing my comments.

1) Section 2.1. Some more information regarding disdrometer data processing are needed. For example, did the Authors apply any kind of pre-processing to disdrometer data such as the elimination of spurious drops using a fall velocity filter (see for example Tokay et al 2001)? There is a minimum number of drops in each considered rainy minute?

2) Section 2.2. Some more information regarding the locations of the devices are needed. For example, which is the distance between radar and disdrometer? Is the disdrometer located along the constant azimuth of the TARA? If yes (or around) which is the height of the first useful TARA bin above the disdrometer?

3) Section 3.1. Please note that also Adirosi et al (2016), among others, have investigated the validity of the gamma assumption to model natural DSD.

4) Section 3.2. It is not clear to me why the Authors used the CF. To estimate mu? Why do not estimate it with MoM as written in the previous sentence? Please clarify

5) I suggest to move section 3.4 before section 3.3

6) Section 4.1. To help the reader can the Authors briefly recall the criterion defined in Gatidis et al (2020) and adopted in the paper? Can the Author provide the percentage of DSD discarded for each event?

7) Section 4.2. What about the mu-lambda relations obtained considering only the "non-gamma DSD"? If (as I guess) it is similar to the one obtained with the whole dataset or considering only "gamma" DSD it means that the assumption that the gamma assumption do not influence the mu-lambda relation is strengthen. Am I correct?

8) Line 235: "previous section" is section 4.1 or 4.2?

9) Section 5.1.1. How do the Authors compute Zh and Zdr from disdrometer data? I guess electromagnetic simulation (such as T-matrix). Please specify

10) Section 5.1.3. I don't understand the need of performing the retrieval considering un-corrected Zh and Zdr. I suggest to eliminate this part and start with the retrieval of the DSD parameter from unbiased Zh and Zdr. This is just a suggestion. The Authors can decide to keep this part but in this case probably a justification is needed.

REFERENCE

Adirosi, E., Volpi, E., Lombardo, F., & Baldini, L. (2016). Raindrop size distribution: Fitting performance of common theoretical models. Advances in Water Resources, 96, 290-305.

Tokay, A., Kruger, A., & Krajewski, W. F. (2001). Comparison of drop size distribution measurements by impact and optical disdrometers. Journal of Applied Meteorology and Climatology, 40(11), 2083-2097.

---

## Author Comment (AC1)

**Response to the reviewers' comments**
* * *
**Reviewer 1**

*The Authors evaluated the effects that different aspects/assumptions can have on the mu-lambda relations and retrieved the mu and lambda parameters from and S-band radar and compared the retrievals with a disdrometer.*
*The papers il well organized and the methodology and results are well described. I suggest the publication on AMT after addressing my comments.*

*1)    Section 2.1. Some more information regarding disdrometer data processing are needed. For example, did the Authors apply any kind of pre-processing to disdrometer data such as the elimination of spurious drops using a fall velocity filter (see for example Tokay et al 2001)? There is a minimum number of drops in each considered rainy minute?*

Answer

Thanks for noticing that. Yes, similar to Gatidis et al. (2020) we applied two criteria to the disdrometer data before the analysis. One in order to ensure all the precipitation is in liquid form, and a second in order to remove suspicious observations.
To clarify, the following sentence was added:
"Similarly to Gatidis et al. (2020), pre-processing is applied to the disdrometer data.
  • Only the liquid type of precipitation was considered for further analysis. All DSDs with observations above the twenty-second diameter class (drop diameters greater than 7mm) were discarded, since they correspond to mixed or solid precipitation.
  • Each DSD should be comprised of at least three different diameter size classes in order to exclude spurious observations not related to rain."

*2)      Section 2.2. Some more information regarding the locations of the devices are needed. For example, which is the distance between radar and disdrometer? Is the disdrometer located along the constant azimuth of the TARA? If yes (or around) which is the height of the first useful TARA bin above the disdrometer?*

Answer

Thank for your comment. We added the following sentences in the text:

"The TARA radar was collocated with additional sensors. This included a Parsivel disdrometer (see Pfitzenmaier et al. 2018, Fig. 1) provided by the Leibniz Institute for Tropospheric Research (TROPOS). For this experiment, the radar antenna elevation angle of TARA was fixed at 45° with constant azimuth. The collected polarimetric radar observables included the reflectivity factor at horizontal polarization ($Z_{hh}$) and differential reflectivity ($Z_{dr}$) at 200 m height (corresponding to the minimum range of TARA)."

*3)      Section 3.1. Please note that also Adirosi et al (2016), among others, have investigated the validity of the gamma assumption to model natural DSD.*

Answer

We have added the following reference in the text:
- Adirosi, E., Volpi, E., Lombardo, F., & Baldini, L. (2016). Raindrop size distribution: Fitting performance of common theoretical models. Advances in Water Resources, 96, 290-305.

*4)      Section 3.2. It is not clear to me why the Authors used the CF. To estimate mu? Why do not estimate it with MoM as written in the previous sentence? Please clarify*

Answer

Thanks for your comment. Even though we mentioned MoM in the manuscript, in reality we did not use it, since the gamma DSD is expressed as

a normalized gamma DSD (Thurai et al. 2014). We re-formulated the text in order to avoid any confusion to the reader:

"The best parameters ($\mu$, $D_m$ and $N_w$) for describing the DSDs measured by the disdrometer are obtained by using normalized parameterization of the Gamma DSD model based on $D_m$ (ratio of $4^{th}$ to $3^{rd}$ order moment). To estimate $\mu$, we first calculate $D_m$ and $N_w$ (directly from the measured DSD spectra). The value of $\mu$ is determined by testing all possible values of $\mu$ between -3 and 15 and choosing the one that minimises the cost function (CF, Eq. 6). Finally, we derive $\Lambda$ through its relationship with $D_m$ and $\mu$ (Eq. 5)."

*5)     I suggest to move section 3.4 before section 3.3*

Answer

Done.

6)     Section 4.1. To help the reader can the Authors briefly recall the criterion defined in Gatidis  et al (2020) and adopted in the paper? Can the Author provide the percentage of DSD  discarded for each event?

Answer

We added the following sentence in the manuscript:
"For this, a filter was applied identical to Gatidis et al. (2020) and only the cases which satisfied the gamma model hypothesis were considered. The adequacy of the gamma model was assessed based on a combination of Kolmogorov–Smirnov goodness-of-fit test and Kullback–Leibler divergence. In total, approximately 40% of the DSDs passed the tests and were accepted. On an event to event basis, that number varies between 36% to 45%."

*7)     Section 4.2. What about the mu-lambda relations obtained considering only the "non-gamma   DSD"? If (as I guess) it is similar to the one obtained with the whole dataset or considering only "gamma" DSD it means that the assumption that the gamma assumption do not influence the mu-lambda relation is strengthen. Am I correct?*

Answer

Actually, this information was already provided in the paper: As stated in the manuscript, α changes from 0.514 to 0.518 and 0.531 respectively and β from 1.339 to 1.328 and 1.312 respectively for the Non Gamma DSDs. Therefore, the acceptance or rejection of the Gamma hypothesis only results in small differences in terms of $\mu$-$\Lambda$ scatterplots. The fact that the relationship remains relatively stable regardless of the gamma DSD assumption, shows that the accuracy of the model assumption does not significantly influence the $\mu$-$\Lambda$ relation itself. However, we should not forget that our dataset consists of relatively similar, stratiform events with light to moderate intensity rain. So, it would be interesting to expand this study to convective events in order to have a more complete picture of the gamma assumption and its influence on the $\mu$-$\Lambda$ relation.

8) *Line 235: "previous section" is section 4.1 or 4.2?*

Answer

The overall $\mu$-$\Lambda$ relationship was introduced and discussed in section 4.1. In order to clarify, we modified that particular sentence as follows:
"Using the overall relationship from Section 4.1 as a reference, the influence of the number concentration on the $\mu$-$\Lambda$ relationship was investigated."

9) *Section 5.1.1. How do the Authors compute Zh and Zdr from disdrometer data? I guess electromagnetic simulation (such as T-matrix). Please specify*

Answer

Indeed, some details were missing. We added the following sentences in order to clarify how $Z_{hh}$ and $Z_{dr}$ were computed from disdrometer data.
"For the sake of the comparison between TARA and Parsivel observables, the radar equivalent reflectivity factor derived from disdrometer data was used as the measured reflectivity factor at horizontal polarization ($Z_{hh,Pars}$). As for the differential reflectivity, using Rayleigh scattering, the calculated radar crosssections of raindrops with equivolume spherical diameter D at horizontal and vertical polarization were used (Eq. 9) for estimating reflectivity at horizontal and vertical polarization, respectively. From those, the differential reflectivity value from Parsivel ($Z_{dr,Pars}$) can be obtained."

*10)    Section 5.1.3. I don't understand the need of performing the retrieval considering un-corrected Zh and Zdr. I suggest to eliminate this part and start with the retrieval of the DSD parameter from unbiased Zh and Zdr. This is just a suggestion. The Authors can decide to keep this part but in this case probably a justification is needed.*

Answer

Thanks for your suggestion. We show the retrievals before and after the correction in order to highlight the importance of the calibration. As we clearly state in the text, the calibration of radar observables ($Z_{hh}$ and $Z_{dr}$) are often overlooked. Hence we think it is really valuable to show both results.

---

## Author Comment (AC2)

**Response to the reviewers' comments**
* * *
**Reviewer 2**

*Abstract*

\*      *Line 4: There is no term used in the statistical gamma family of distributions that has the term "constrained gamma". The mu-lambda relation is an empirically derived based on measured DSDs. Since the measured DSDs are statistical (i.e. the parameters such as Dm can be treated as statistical) the mu-lambda is not a deterministic relation.*

Answer

To clarify, the following sentence has been added to the paper:
"When an empirical relation between shape and scale parameters is used the model is often called constrained-gamma. Note that the term "constrained-gamma" denote a gamma DSD model in which the shape and rate parameters are linked by a deterministic function. Mathematically, this is equivalent to reducing the number of free parameters from three to two, which is convenient in radar-based DSD retrievals. However, the uncertainty related to estimating μ and Λ based on observed DSD spectra remains. Hence the constrained gamma DSD model and all its associated moments still remains stochastic in nature."

\*      *Line 12: Sentence beginning 'The most difficult ..' This is true of all retrievals of the DSD and R. It is not surprising that NT which is 0th moment of the DSD cannot be estimated accurately using higher order moments like Z=f(M6) and Dm=M4/M3.*

Answer

Yes, indeed, it is intrinsically hard to retrieve low order moments such as $N_T$ from higher order moments such as Z. That, combined with the fact that there is some error propagation in the retrieval procedure (i.e., $N_T$ is the last parameter to be retrieved) makes it very challenging to get accurate estimates of $N_T$.

*       *Abstract, Last sentence: this increase in correlation from 0.12 to 0.24 is not a meaningful increase...the scatter still looks "random"*

Answer

Thanks for your comment. We modified the sentence as follows:
"After careful data filtering and removal of problematic $Z_{hh}/Z_{dr}$ pairs, the correlation coefficient for the retrieved $N_T$ values remained low, only slightly increasing from 0.12 into 0.24."

*       *Line 33: Surely by now the entire DSD community is aware that N0-mu relation is not physical.*

Answer

Noted. But it does not hurt to repeat it and provides some useful context to young career scientists who just started working on the topic. It may also be useful to people who are not very familiar with the theory behind drop size distributions.

*       *Line 46: I do not agree that calibration offsets in Zh and Zdr are often overlooked. The US Nexrad system has done extensive work to reduce the uncertainty of Zdr to within -+0.1 dB. To this, one can add the German DWD, and MeteoFrance as well.*

Answer

Yes, we know that there are operational radars (like the ones you mentioned above) for which there is an in-depth procedure to monitor and correct

calibration issues. However, unfortunately, this is not the case everywhere, especially for research radars. Our study highlights the importance of this issue, without minimizing the good work done by other researchers, institutes and agencies. In order to convey the right meaning, we slightly modified the corresponding text:

"Finally, one last issue that tends to be overlooked is that radar measurements are likely to contain systematic errors in the form of calibration offsets in $Z_{hh}$ and $Z_{dr}$. A possible error in the latter could induce large biases in the retrieved DSDs, especially in light rain with low $Z_{dr}$ and small signal to noise ratio. Several operational polarimetric weather radar networks such as the US Nexrad (Hubbert and Pratte, 2007) and the German DWD network (Frech and Hubbert, 2020) have already devoted extensive efforts toward mitigating these calibration issues. However, achieving and maintaining good calibration over time for research radars remains challenging."

*       Line 71: The instrument does not possess the resolution to measure the drizzle and very small drops. This is also termed as truncation of the DSD and the shape factor will be biased to strongly positive values with convex down shape at the small drop end.*

Answer

Thank you for your comment. As we stated in text, we are perfectly aware of the limitations of the Parsivel. The modified sentence (please see below) now clearly mentions that Parsivel has difficulties measuring small drops:

"The working principle, strengths and limitations of the PARSIVEL[2] have already been discussed in great depth in previous studies and will not be part of this study (Löffler-Mang and Joss, 2000; Tokay et al., 2014; Battaglia et al., 2010; Thurai et al., 2011, Raupach and Berne 2015;). For example, the Parsivel is susceptible to errors in the lower drop diameter range which can affect the DSD shape and number concentrations. However, no efforts have been done to try to correct for these issues within the context of this study."

*       Line 85: "comparable" is not the correct description.... you are only sampling in time to get 30 s sampling.*

Answer

Thank you for the comment. Yes, strictly speaking they are not comparable. However, since our only option is to compare data from different sensors which have different specifications, we should at least try to first adjust them in a way in order to make them comparable to each other in a sort of a way. After we down-sampled TARA's $Z_{hh}$ and $Z_{dr}$ measurements over successive 30 s sampling intervals, even though the sensors are no similar, we could say that their data are kind of comparable.

*       *Line 98: fig 1 does not appear to have a clear melting layer....what is mean by clear? the vertical streaks of Z above the BB indicates vertical air motion.*

&

*       *Line 112: the BB does not look steady, rather the vertical streaks in Z well above the BB depict some vertical air motion.*

Answer

Thanks for your comment. Yes, indeed vertical streaks of reflectivity above the bright band indicate some vertical air motion. However, the classification into stratiform and convective should not be taken too strictly as events are likely to contain a mixture of different rain types. To clarify this point, the text has been modified to:

"1. Each event must consist of predominantly stratiform rain and exhibit a well-defined melting layer signal in the radar data."

*       *Eq. 1: the use of NT was introduced by Chandrasekar and Bringi to emphasie that NT = 0th moment =total number density which makes this form similar to what statisticians would use.*

Answer

Noted. We added the following reference in the text:
- Bringi, V. N., and V. Chandrasekar, 2001: Polarimetric Doppler Weather Radar: Principles and Applications. Cambridge Uni- versity Press, 636 pp.

*       *Line 154: "empirical" or "statistical"?*

Answer

Done. Empirical relationship.

*       *Eq. 7: is there any physical basis for this power law?*

Answer

Thank you for your comment. We re-arranged Section 3 and now Subsection 3.4 is before Subsection 3.3 where we clarify the reason behind our model choice. As stated in the manuscript the power law model is easier to physically justified rather than a parabola.

*       *Line 163: Dmax is approximately 3*Dm...see Carey and Petersen*

Answer

Thanks for the comment. Yes, indeed, we modified the sentence as follows: "where $\sigma_{hh}$ (mm$^2$) and $\sigma_{vv}$ (mm$^2$) are the copolar radar cross-sections of raindrops with equivolume spherical diameter D, at horizontal and vertical polarization, respectively, and $D_{max}$ (mm) is a reasonable maximum drop diameter (e.g., 7 mm in our case). In the literature several studies tried to link $D_{max}$ with $D_0$ such as Ulbrich and Atlas (1984), who concluded that $D_{max} / D_0 >$ 2.5 is what is typically observed in natural rainfall, or Carey and Petersen (2015) who recommended using $D_{max} = 3 * D_0$."

*       *Line 195: The critical aspect is that Parsivel cannot measure the drizzle or smalll drops with sufficient resolution causing truncation. This causes Dm to increase as well as the decrease in the the spectral width sigma... casing mu to decease.*

Answer

Yes, this is true. We added a sentence to highlight the effect of truncation to the DSD shape itself.

"Critical aspects that were investigated are whether the μ-Λ relation remains stable with respect to different sampling resolutions, drop number concentrations, types of stratiform rain events or the validity of the gamma DSD hypothesis itself. At the same time, one has to keep in mind that the limitation of the Parsivel in terms of detection of small droplets might lead to overestimated $D_m$ and μ values, since the width of the distribution will be underestimated."

*       *Also, the stability of mu-lambda relation itself is not in question since it can be stable for the wrong reason.*

Answer

This does not make any sense. Why would the μ-Λ be stable for the wrong reason? And what is the link with the previous paragraph/comment?

*       *Line 235: The NT is the same as M0 ie the total number density. It is not possible to estimate it from the higher order moments such as Nw or Dm. In fact the variability in NT of the DSD is larger than that of Dm or mu. This is termed as number controlled DSDs.*

Answer

Thanks for the comment. The fact that DSDs are predominantly number or size-controlled does not really play a crucial role in the retrieval algorithm itself. However, it is possible that the rainfall regime (i.e., number vs size-controlled DSDs) and associated scaling laws could influence the stability of the μ-Λ relationship. This is interesting but would have to be investigated in a separate paper, as it is clearly outside the scope of this study and would require new data for convective events as well. An additional sentence has been added to the text to clarify this point:

"It would be interesting to investigate whether the events for which the DSD is predominantly number controlled lead to more/less stable μ-Λ relationships than events with size-controlled DSDs."

*    *Last sentence in 5.1.3: this is known as the point-to-area or non-uniform beam filling problem. This is very well known and has been addressed by several publications*

Answer

Thank you for your comment. We added the following reference in the text:

- Ryzhkov, A. V. (2007). The Impact of Beam Broadening on the Quality of Radar Polarimetric Data, Journal of Atmospheric and Oceanic Technology, 24(5), 729-744.
- S. L. Durden and S. Tanelli, "Predicted Effects of Nonuniform Beam Filling on GPM Radar Data," in IEEE Geoscience and Remote Sensing Letters, vol. 5, no. 2, pp. 308-310, April 2008, doi: 10.1109/LGRS.2008.916068.

*    *Last sentence, 5.2: This is not surprising since NT is the M0th moment whereas Nw, Dm are of much higher order.*

Answer

Yes, indeed. See our response to previous, similar comments.

*    *Line 405: no surprise here...unless one can measure M1, M2, there is no way to improve the estimate NT.*

Answer

Thank you for the comment but we do not really agree with this. The estimation of $N_T$ is possible without M1 and M2. If we assume that the DSDs are perfectly gamma, that $Z_{hh}$ and $Z_{dr}$ are perfectly calibrated, and that the μ-Λ relationship is valid, the $N_T$ could in principle be estimated. The problem is that we have large measurement and modeling uncertainty. But there are plenty of ways to improve, for example by applying bias corrections (over time), and also, potentially, by adapting the μ-Λ relationship over time.

*     *Line 447: The method of improving the correlation coeff especially for NT does not improve at al ...the corr~ 0.*

Answer

Done. See our response to previous, similar comments.

---

## Author Comment (AC3)

**Response to the reviewers' comments**
* * *
**Reviewer 3**

*This manuscript estimates variability of Lamba-mu relations of the assumed gamma-function DSD in observed liquid precipitation. The results obtained in this study may be useful for better understanding of uncertainties in these relations. I recommend a major revision of the manuscript having in mind comments below.*

*Main comments.*

*\*      The authors should clarify their retrieval method described in section 3.3. They describe how they estimate mu (steps 2 and 3). How the corresponding Lambda value is then obtained? They state that they impose a fixed Lambda – mu relation with fixed coefficients (i.e., relation (7)). If they use this fixed relation then how different prefactors and exponents (alpha and beta in Table 1) are obtained?*

Answer

Thank you for your comment. In Section 3.3 we describe how the DSD parameters were retrieved using a combination of radar observations and a fixed relationship between μ and Λ. However, in Table 1 we do not follow these steps since we deal with disdrometer data only, so μ and Λ are coming directly from the observations (no need for retrievals). We modified Section 3.3 accordingly in order to make that more clear to the reader.

*\*      Please provide a better description of the geometry of measurements. What are relative locations of the disdrometer and the radar? At what heights radar*

*measurements are made? Is the disdrometer directly below the radar resolution volume? In other words, what are horizontal and vertical distance separations between the radar and disdrometer.*

Answer

Thank for your comment. We added the following sentences in the text:
"The TARA radar was collocated with additional sensors. This included a Parsivel disdrometer (see Pfitzenmaier et al. 2018, Fig. 1) provided by the Leibniz Institute for Tropospheric Research (TROPOS). For this experiment, the radar antenna elevation angle of TARA was fixed at 45° with constant azimuth. The collected polarimetric radar observables included the reflectivity factor at horizontal polarization ($Z_{hh}$) and differential reflectivity ($Z_{dr}$) at 200 m height (corresponding to the minimum range of TARA)."

*\*        Are coefficients in (7) simple mean values or are they some kind of weighted mean values? (for example, weighted by event durations, etc.).*

Answer

They are simple mean values across the 7 selected events without any weights.

*\*        Equations (1) through (5) assume untruncated distributions. Do you have any estimates how truncation to Dmax in (9) and (10) would affect the results? I assume that this effect is mu-dependent.*

Answer

We did not explicitly investigate this issue because the drop diameters considered in this study were rather small. Therefore, it is reasonable to assume that the truncation with $D_{max}$ does not substantially affect the results. Similarly, the choice of the actual value for $D_{max}$ (e.g., 6 or 7 mm) is very unlikely to change the μ-Λ relationships and our conclusions.
In general, we are perfectly aware of the limitations of the Parsivel in terms of detection of small droplets which could lead to overestimated $D_m$ and μ values, since the width of the distribution will be underestimated.

*       Line 164: Eq.(3) from Unal (2015) shows only horizontal polarization backscatter cross section. Do you account for the elevation angle for the vertical polarization cross section? What were assumed drop orientations?*

Answer

Yes, we account for the elevation angle of $45^{o}$ to calculate the radar cross section at vertical polarization. For the raindrop canting angle distribution, a Fisher distribution symmetric around $0^{o}$ with a width parameterized by κ being 30 is used.

*       What are your estimates of uncertainties in the Lambda-mu estimates? Given the retrieval/measurement uncertainties, are the results for different events shown in Fig.3 really statistically different?*

Answer

We did not estimate the uncertainty explicitly but it is quite clear that at such small time scales, uncertainties on μ and Λ can be substantial. There is no real need to calculate these uncertainties because, as we already highlighted in the text, apart from events 2 and 6, for which the overall relations are obviously different, the rest of the events had very similar μ-Λ relationships that were well within the expected uncertainty range for μ and Λ.

*       The correlation coefficients of 0.12 - 0.24 for retrieved Nt (as mentioned in the abstract) actually indicate no reliable correlation.*

Answer

Thanks for your comment. We modified the sentence as follows:
"After careful data filtering and removal of problematic $Z_{hh}/Z_{dr}$ pairs, the correlation coefficient for the retrieved $N_T$ values remained low, only slightly increasing from 0.12 into 0.24."

*I suggest calculating a power-law correlation coefficient between Lambda and mu for each event and also RMSD between individual Lambda – mu points and the best fit. Showing these statistical metrics in in Table 1 would be beneficial.*

Answer

Thanks for your comment. We included the correlation coefficient and RMSD in Table 1.

*Why not to use lower elevation angle for radar measurements to increase ZDR?*

Answer

Indeed, a lower elevation angle would increase the value of $Z_{dr}$. However, during the ACCEPT campaign, only the 45° elevation was considered, which was the optimal choice in order to combine polarimetric and Doppler spectra information and perform other, microphysical studies.

*Minor comments*

*Since you use binned DSD information, you should probably use summations in (9) and (10) rather than integrals.*

Answer

Only the DSD data derived from the Parsivel are binned. Equations 9 and 10 are used for the DSD retrievals from the radar data, which are not binned. That is why an integral is the correct mathematical expression.

*Equations (7) and (11) are repetitive.*

Answer

Thank you for your comment. Since we moved section 3.4 before 3.3 to address another comment, we should keep both equations 7 and 8 in order to avoid any confusion for the reader.

*       *The first line after (9): here capital Lambda size parameter and small lambda - wavelength are mixed up.*

> Answer

> Thanks for spotting this mistake! We made the necessary changes.

*       *Add Zdr frame to Fig. 2.*

> Answer

> Thank you for the suggestion, but we do not think that this is necessary. The purpose of Figure 1 and Figure 2 is to help visualize the events and to compare some basic DSD moments such as radar equivalent reflectivity factor, rain rate etc. between the Parsivel and TARA.

*       *Line 296 says: see Section 3a, but there is no section 3a in the paper. Is it 3.1 ? Also you are referring to section 3c in line 340 (and in other parts of the paper), but it probably should be section 3.3. Check the entire manuscript for consistency in referencing different sections.*

> Answer

> Done.

*       *Are sigma' and sigma in lines 302-304 the same parameter?*

> Answer

> Yes. Actually $\sigma'$ is the new $\sigma$ (mass spectrum standard deviation).